# Association of C-reactive protein with future development of diabetes: a population-based 7-year cohort study among Norwegian adults aged 30 and older in the Tromsø Study 2007–2016

Kit I Tong,[1] Laila Arnesdatter Hopstock,[2] Sarah Cook [3,4]

¹Princess Margaret Cancer Centre, University Health Network, Toronto, Ontario, Canada
²Department of Health and Care Sciences, UiT The Arctic University of Norway, Tromsø, Norway
³School of Public Health, Imperial College London, London, UK
⁴Faculty of Epidemiology and Population Health, London School of Hygiene & Tropical Medicine, London, UK

**Correspondence to**
Dr Sarah Cook;
sarah.cook@imperial.ac.uk and
Dr Kit I Tong;
kit.tong@uhn.ca

## ABSTRACT

**Objectives** The extent to which observed associations between high-sensitivity C-reactive protein (hs-CRP) and incident diabetes are explained by obesity and hypertension remains unclear. This study aimed to investigate the association of hs-CRP with developing diabetes in a Norwegian general population sample.

**Design** A cohort study using two population-based surveys of the Tromsø Study: the sixth survey Tromsø6 (2007–2008) as baseline and the seventh survey Tromsø7 (2015–2016) at follow-up.

**Setting** Tromsø municipality of Norway, a country with increasing proportion of older adults and a high prevalence of overweight, obesity and hypertension.

**Participants** 8067 women and men without diabetes, aged 30–87 years, at baseline Tromsø6 who subsequently also participated in Tromsø7.

**Outcome measures** Diabetes defined by self-reported diabetes, diabetes medication use and/or HbA1c≥6.5% (≥48 mmol/mol) was modelled by logistic regression for the association with baseline hs-CRP, either stratified into three quantiles or as continuous variable, adjusted for demographic factors, behavioural and cardiovascular risk factors, lipid-lowering medication use, and hypertension. Interactions by sex, body mass index (BMI), hypertension or abdominal obesity were assessed by adding interaction terms in the fully adjusted model.

**Results** There were 320 (4.0%) diabetes cases after 7 years. After multivariable adjustment including obesity and hypertension, individuals in the highest hs-CRP tertile 3 had 73% higher odds of developing diabetes (OR 1.73; p=0.004; 95% CI 1.20 to 2.49) when compared with the lowest tertile or 28% higher odds of incidence per one-log of hs-CRP increment (OR 1.28; p=0.003; 95% CI 1.09 to 1.50). There was no evidence for interaction between hs-CRP and sex, hypertension, BMI or abdominal obesity.

**Conclusions** Raised hs-CRP was associated with future diabetes development in a Norwegian adult population sample. The CRP-diabetes association could not be fully explained by obesity or hypertension.

## INTRODUCTION

Diabetes is a diverse metabolic disorder primarily characterised by impaired glucose

### STRENGTHS AND LIMITATIONS OF THIS STUDY

⇒ The 7-year population-based cohort study design allowed us to investigate the temporal association between high-sensitivity C-reactive protein and future diabetes development.

⇒ The study used a large overall cohort sample size with high-quality, high participation rate and a low level (<11%) of missing data, which garners high internal validity.

⇒ A number of potential confounders were taken into account including medication use of lipid-lowering drugs, which were not included in other previous studies.

⇒ Definition of the baseline and outcome future diabetes were based partly on self-reported diagnosis and self-reported diabetes medication use although the diabetes case definition was additionally based on the more robust HbA1c laboratory-based blood test to improve both sensitivity and specificity.

⇒ The study focused on a sample of the Norwegian general population from one municipality in Northern Norway and hence the conclusions may not be generalisable beyond this region.

tolerance and hyperglycaemia resulting from insulin deficiency or resistance. In 2019, there were approximately 463 million people with diabetes worldwide and the number of patients is constantly increasing.[1] The growing hyperglycaemia-related deaths, healthcare burden and the associated socioeconomic outcomes are pressing issues. Identification of modifiable risk factors and predictive biomarkers for diabetes becomes important to offer opportunities for early prevention and intervention.

Due to the conflicting broad definitions of type 1 and type 2 diabetes, there was a recent proposal for a β-cell-centric classification system of diabetes that encompasses interactions between β-cells function and abundance

with genetics, insulin resistance, environmental factors, immune dysregulation and inflammation, which underlie a wide spectrum of hyperglycaemic phenotypes found in patients with diabetes.[2] Standardisation of laboratory-based tests such as those that measure C-peptide, low-grade inflammation, β-cells mass, insulin resistance and β-cells autoantibodies may facilitate better subtype diagnosis and subsequent precision care of diabetes.[2 3]

While disease progression of type 1 diabetes involves inflammation of the islet cells leading to ultimate massive β-cells loss and clinical diabetes,[4 5] systemic inflammation is considered as one of the mechanisms mediating the pathogenesis of type 2 diabetes.[4] Inflammation is a cellular defence system against infections and is an essential mechanism for normal tissue homoeostasis.[6] Inflammatory cells produce cytokines leading to the production of acute phase proteins in the liver to promote tissue repair.[7] However, when timely resolution reinstating cellular homoeostasis fails, the resulting chronic inflammation can also lead to tissue damage.[8] Given the emerging role of inflammation in the pathophysiology of diabetes and the associated metabolic disorders, there is increasing interest in the development of anti-inflammatory therapeutics to improve both the prevention and the management of diabetes with numerous clinical trials showing various promising effects on patients with type 1 and type 2 diabetes.[4 9 10] Several risk factors for diabetes—ageing, obesity, smoking and physical inactivity—have been found to be positively associated with low-grade inflammation.[11–14] Excessive energy intake and sedentary lifestyle contribute to fat accumulation in tissues including muscles, liver and pancreas, which may trigger chronic low-grade inflammation through adipokine-induced production of proinflammatory cytokines and chemokines, ultimately leading to insulin resistance and β-cell dysfunction.[10] Indeed, the elevation of acute-phase proteins such as C-reactive protein (CRP), a marker for systemic inflammation, has previously been shown to be associated with development of type 1 and type 2 diabetes.[15 16]

Demographic factors such as sex,[17 18] age[11 19] and education[20]; lifestyle or behaviour risk factors for diabetes and/or CRP levels such as alcohol consumption,[21 22] daily smoking[13 23] and physical inactivity[14 24]; clinical risk factors such as high body mass index (BMI),[12 25] abdominal obesity,[12 26 27] non-HDL cholesterol,[28 29] medication use of lipid-lowering drug,[30 31] and the comorbidity hypertension are potential confounders that have been indicated associated with CRP levels and/or with diabetes incidences[32]; but not mediating the pathway between the association of CRP and diabetes incidences (figure 1).

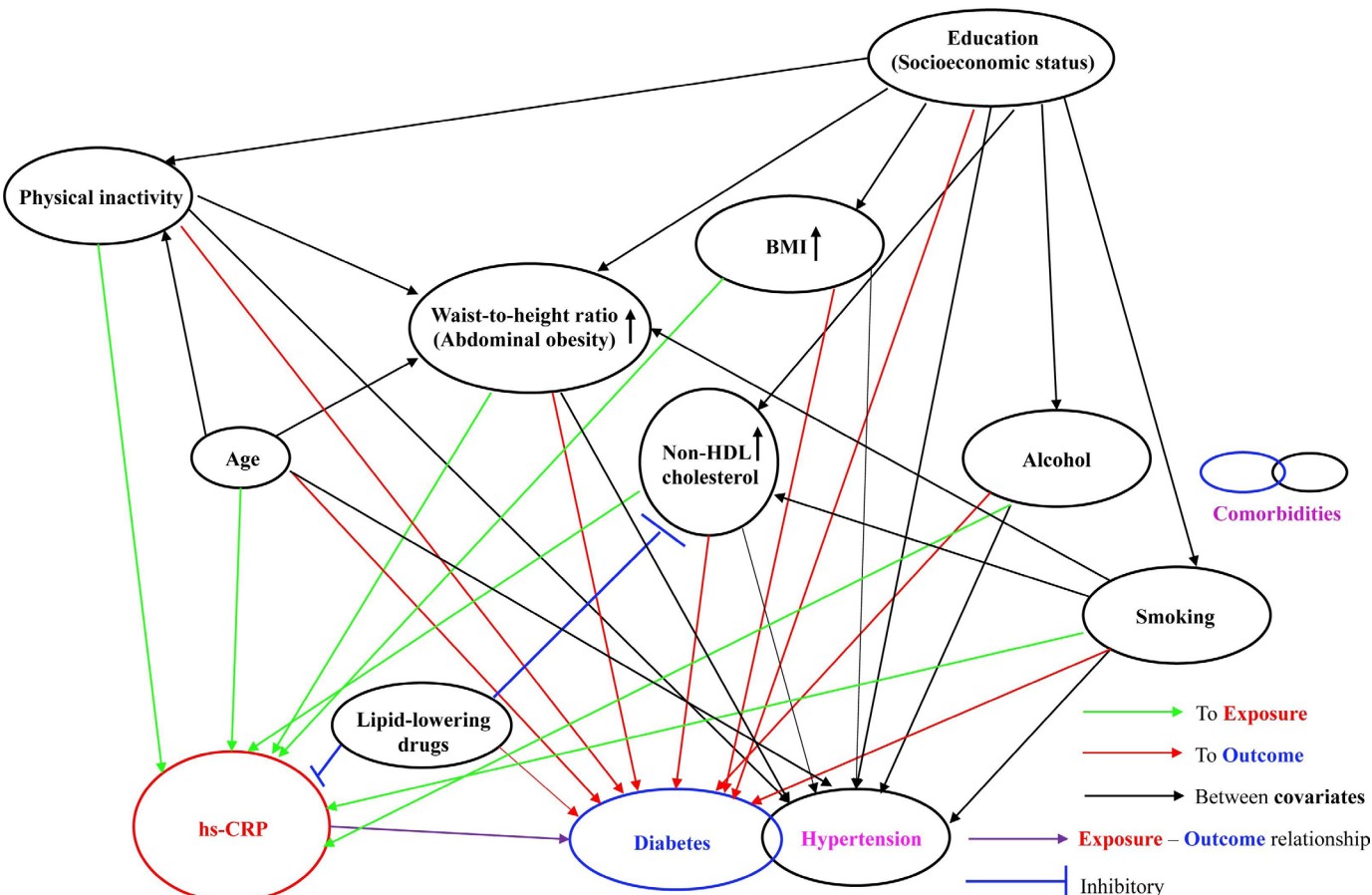

**Figure 1** Conceptual framework. BMI, body mass index; HDL, high-density lipoprotein; hs-CRP, high-sensitivity C-reactive protein.

There are conflicting conclusions in the literature on whether the association between high-sensitivity CRP (hs-CRP) and diabetes development is modified by sex and whether the association is truly independent of hypertension,[15 33–35] general obesity[36–38] and abdominal obesity.[33 39] Evidences of interaction were also evident in some studies between CRP and sex,[34 40] BMI,[15 38] hypertension[15 41] or with waist circumference,[38] while other studies, however, did not observe evidence of interaction between hs-CRP with sex nor with BMI.[42 43]

Lipid-lowering drugs such as statins, fibrates, nicotinic acid and some other lipid-modifying agents have both a lipid-lowering capacity and anti-inflammatory effect.[44] The use of these medications has been found able to lower hs-CRP levels[30]; while, conversely, the recipients of the drugs are inherently at risk of diabetes development due to indications for the medication use to reduce blood cholesterol and triglyceride levels.[45 46] Some, such as statins, are associated with elevated risk of new onset of diabetes while nicotinic acid is associated with impaired blood glucose control suggesting potential drug-induced side effects.[31] Some previous studies were not able to or did not adjust for the parameter of medication use leading to the questions of potential residual confounding.[15 33 36]

According to Norwegian Institute of Public Health, there was high prevalence of overweight and obesity combined in Norway.[47] Using a recent revised consensus definition of metabolism syndrome from the harmonised Adult Treatment Panel-III criteria,[48] a recent repeated large cohort cross-sectional study of the rural Northern Norway has found that the prevalence of age-standardised metabolic syndrome (MetS) has progressed from 31.2% in 2003–2004 (n=6550) to 35.6% in 2012–2014 (n=6004) with concomitant increase in the mean MetS severity, which was believed to be driven by the marked increase in abdominal obesity among the study population over the years.[49]

Population-based studies assessing CRP as a marker for future diabetes incidences have not been reported for the Scandinavian countries. We aimed to investigate the association of hs-CRP on the development of diabetes after an interval of 7 years among adults in Norway, a Nordic country with an increasing proportion of middle-aged and older adults and with a high prevalence of overweight, obesity and hypertension.

## METHODS
### Setting
The Tromsø Study is a population-based health study in the Tromsø municipality, Norway.[50] Participants were invited from the whole birth cohorts and random sample of the Tromsø residents. Data collection includes questionnaires, biological samples and clinical examinations.

This study involved data from two surveys of the Tromsø Study: the sixth survey of the Tromsø Study (Tromsø6) 2007–2008 as baseline and the seventh survey of the

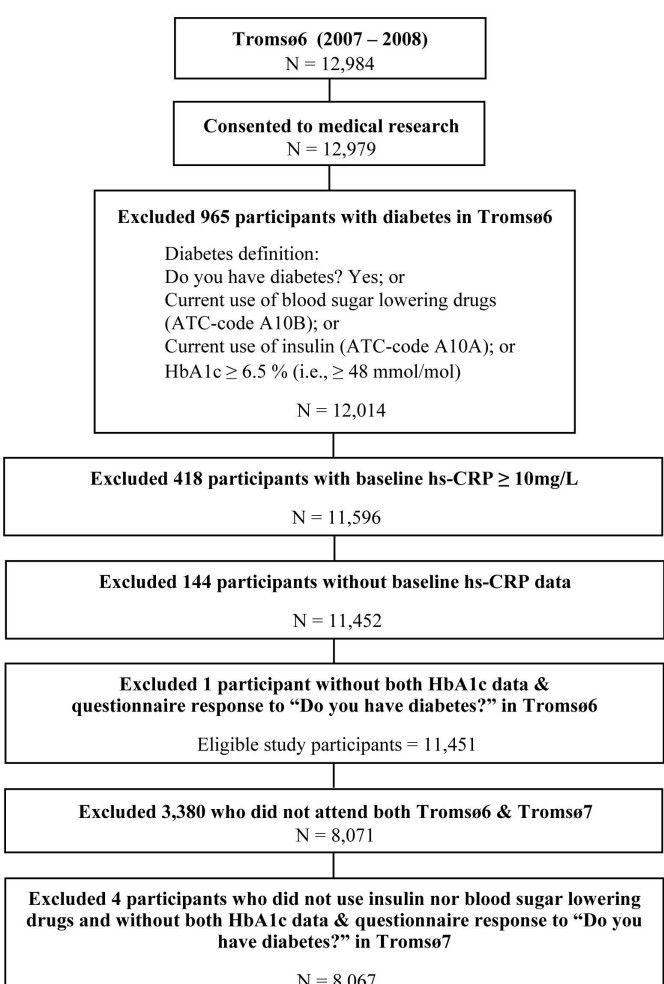

**Figure 2** Study sample inclusion flow chart of the Tromsø Study 2007–2016. ATC, Anatomical Therapeutic Chemical; HbA1c, glycated haemoglobin; hs-CRP, high-sensitivity C-reactive protein.

Tromsø Study (Tromsø7) 2015–2016 as the follow-up time point after a 7-year interval.

In Tromsø6 2007–2008, all residents aged 40–42 and 60–87 years (n=12 578), a 10% random sample of residents aged 30–39 years (n=1056), a 40% random sample of residents aged 43–59 years (n=5787) and all previous participants who attended Tromsø4 second visit (if not already included in the three groups above) (n=341) were invited.[51] In total, 12 984 women and men attended Tromsø6 with 66% participation (figure 2).

In Tromsø7 2015–2016, all residents aged 40 years and older were invited (N=32 591).[52] In total, 21 083 women and men attended Tromsø7 with 65% participation (figure 2).

### Study design
A cohort study used two population-based surveys of the Tromsø Study: the sixth survey Tromsø6 (2007–2008) as baseline and the seventh survey Tromsø7 (2015–2016) at follow-up after a 7-year interval. The association of the baseline hs-CRP, expressed as three quantiles, at Tromsø6 with future diabetes cases at Tromsø7 for the participants

who did not have baseline diabetes at Tromsø6 was examined by logistic regression adjusted for a number of a priori potential confounding factors and then further tested for potential effect modifications by likelihood-ratio tests (LRTs). The CRP–diabetes association was further tested in the sensitivity analysis by repeating multivariable logistic regression using the baseline log hs-CRP, at Tromsø6, expressed as continuous independent variable.

### Study sample

Figure 2 depicts the flow chart for the selection of the study sample. In brief, participants without prevalent diabetes in Tromsø6 who had also attended Tromsø7 were included in this analysis.

In addition, individuals with baseline hs-CRP≥10 mg/L, an indication of acute inflammatory condition at the time of examination,[53–57] would be excluded in the final analysis to mitigate the risk of measurement error. As acute inflammation may resolve over a short period of time and may not necessarily transit into chronic inflammatory condition.[58] A previous study involving 5111 men aged 48–77 in the Oslo Study showed that individuals with health conditions of osteoporosis, asthma, chronic bronchitis/emphysema, diabetes or myocardial infarction had a mean CRP level of 6.53 mg/L, 5.01 mg/L, 4.42 mg/L, 4.53 mg/L and 4.27 mg/L, respectively.[59] These values were much less than hs-CRP of 10 mg/L.

In details, 12 984 participants attended Tromsø6. We excluded 5 participants who did not consent to medical research; 965 participants with baseline diabetes; 418 participants with baseline hs-CRP≥10 mg/L; 144 participants without baseline hs-CRP data and 1 participant without HbA1c nor self-reported diabetes diagnosis data with negative response to the use of insulin and blood sugar lowering drugs for determining baseline diabetes status. Among the 11 451 participants without baseline diabetes in Tromsø6 with hs-CRP<10 mg/L and valid measurements, a total of 8071 also attended Tromsø7. Additionally, four participants were excluded due to inadequate information in Tromsø7 to determine diabetes status. Hence, 8067 (3,704 men, 45.9%; 4363 women, 54.1%) participants without baseline diabetes nor acute inflammation in Tromsø6 aged 30–87 years were included in the final analysis (figure 2).

Criteria defining diabetes condition are as described below in the 'Outcome definition (diabetes)' section.

### Data collection procedures

Data obtained by questionnaires, laboratory analyses and clinical examinations have been described previously.[51 52 60] Briefly, blood samples without mandatory fasting were collected by EDTA anticoagulation tubes to measure HbA1c levels by high-performance liquid chromatography with Variant II (Bio-Rad Laboratories, Hercules, California, USA) in Tromsø6 and by Tosoh G8 (Tosoh Bioscience, San Francisco, USA) in Tromsø7. Serum hs-CRP, total cholesterol and high-density lipoprotein

(HDL) cholesterol were measured with immunoturbidimetric and enzymatic colorimetric methods, respectively, by Modal PPE autoanalyzer (Roche Diagnostics Norway AS) in Tromsø6. Height in centimetres (cm) and weight in kilograms (kg) were measured by a Jenix DS 102 stadiometer (Dong Sahn Jenix, Seoul, Korea). Waist circumference was measured in cm by a Seca measuring tape at the level of the umbilicus. Three separate readings of systolic and diastolic blood pressure were taken, using a Dinamap Pro care 300 monitor (GE Healthcare, Norway), with 1 min of intervals between each reading after a 2 min seated rest. The mean systolic and diastolic blood pressure of readings 2 and 3 were used in this study.

### Variables in the analyses

#### Exposure (hs-CRP)

The baseline hs-CRP at Tromsø6, stratified into three quantiles of ascending values, was used as laboratory-based proxy measurement for chronic inflammation to deduce its impact on the odds of developing diabetes over 7 years in Tromsø7. Additionally, in the sensitivity analysis, hs-CRP was first log transformed due to its nonnormally and skewed distribution. The log transformed hs-CRP was then used as a continuous independent variable to reanalyse the regression model.

#### Outcome definition (diabetes)

Known and unknown incident diabetes cases at Tromsø7 were identified by glycated haemoglobin (HbA1c) ≥6.5% (ie, ≥48 mmol/mol),[60] and/or self-reported diabetes diagnosis ('Do you, or have you had diabetes?'), and/or self-reported use of blood sugar lowering (Anatomical Therapeutic Chemical (ATC) code A10B) and/or insulin-containing medication (ATC code A10A) for treating symptoms.[61] Self-reported previous diabetes only was not defined as diabetes.

#### Potential confounders

As described in the conceptual framework in figure 1 and in the introduction, previous studies suggested that the following variables are potential confounders that have been indicated associated with CRP levels and/or with diabetes incidences,[11–14 17–32] but not mediating the pathway between the association of CRP and diabetes incidence: they include demographic factors (age, sex and education); behavioural risk factors (alcohol consumption frequency, daily smoking status and physical activity); cardiovascular risk factors (BMI, abdominal obesity and non-high-density lipoprotein (non-HDL) cholesterol); use of lipid-lowering medications (ATC code C10)[61] and comorbid hypertension.

#### Potential effect modifiers

Previous studies suggested that sex, BMI category, abdominal obesity or hypertension may interact with hs-CRP although conflicting results have been found from other studies.[15 34 38 40–43]

### Definition of the variables (confounders and/or effect modifiers)

► Age was divided into 5-year groups for the individuals aged 40–74 years. Due to data sparsity, the youngest and the eldest individuals were categorised into 10-year (30–39 years) and 12-year (75–87 years) age groups.

► Self-reported education, as a proxy for socioeconomic status, was defined by five levels (primary/ secondary school, 1–2 years senior high school, high school diploma, <4 years university or ≥4 years university).

► Alcohol consumption frequency was defined by five levels (never drinker, monthly or less frequent drinker, 2–4 times per month, 2–3 times per week or ≥4 times per week).

► Daily smoking was defined as current, previous or never.

► Physical activity was defined from self-report as sedentary (reading, watching televisionor other sedentary activity), low (walking, cycling or other forms of exercise at least 4 hours a week), moderate (recreational sports, heavy gardening, etc at least 4 hours a week) or vigorous (hard training or sports competitions, regularly several times a week).[51]

► BMI was categorised as <25 kg/m$^2$, 25–29.9 kg/m$^2$ or ≥30 kg/m$^2$.[47]

► Abdominal obesity was categorised as abdominal obese when waist-to-height ratio ≥0.5. It was suggested by previous systematic review and meta-analysis studies that a waist-to-height ratio cut-off of 0.5 is suitable for adults of both sex and different ethnic backgrounds.[27 62]

► Non-HDL cholesterol was derived by negating HDL cholesterol from total serum cholesterol.

► Hypertension was defined by systolic blood pressure ≥140 mm Hg and/or diastolic blood pressure ≥90 mm Hg,[63] and/or self-reported use of blood pressure lowering drugs (ATC code C02, antihypertensives; C03, diuretics; C07, beta blocking agents; C08, calcium channel blockers; C09, renin–angiotensin acting agents) for treatment.[61]

### Statistical analysis

The hs-CRP levels of the study participants (table 1) were first stratified into three quantiles, 0.14–0.77 mg/L (tertile 1), 0.78–1.69 mg/L (tertile 2) and 1.70–9.98 mg/L (tertile 3) to conduct most of the analyses described below unless otherwise specified.

#### Age-standardised cumulative diabetes incidence

In figure 3, age-standardised cumulative diabetes incidence across baseline hs-CRP tertiles, overall or sex-specific, was calculated using direct standardisation to the 2013 European Standard Population to examine any dosage effect of hs-CRP exposure on outcome diabetes cases.[64]

#### Baseline characteristics

In table 1, the descriptive statistics of the baseline characteristics for the sample population across the three quantiles of baseline hs-CRP was examined:

By Pearson's $\chi^2$ test for categorical variables (sex, age, BMI category, hypertension, blood pressure lowering drugs, abdominal obesity, lipid-lowering drugs, alcohol consumption frequency, daily smoking, physical activity, education).

By Kruskal-Wallis test for continuous variables with non-normal distribution (hs-CRP).

By one-way analysis of variance for continuous variables with normal distribution (waist-to-height ratio, non-HDL cholesterol, systolic blood pressure, diastolic blood pressure).

#### Logistic regression models

In table 2, crude model was built to examine the association of baseline hs-CRP tertiles with diabetes incidence in Tromsø7. The model was then multivariable adjusted for a set of a priori determined covariates (sex, age, BMI, hypertension, abdominal obesity, non-HDL cholesterol, lipid-lowering drug use, alcohol consumption frequency, daily smoking status, physical activity and education) to correct for any potential confounding effects on the CRP-diabetes association.

#### Evaluation of potential effect modification

There were conflicting conclusions in previous studies on whether CRP–diabetes association is effect modified by sex, BMI, abdominal obesity or hypertension.[15 34 38 40–43] In table 2, we examined the potential effect modifiers by adding interaction term for sex (hs-CRP tertile×sex), anthropometric factors (hs-CRP tertile×BMI or hs-CRP tertile×abdominal obesity), or hypertension (hs-CRP tertile×hypertension) separately to the final multivariable logistic regression model and tested the model by LRT. The p value of the LRT was used to evaluate whether the addition of any interaction term constitutes a better model. The null hypothesis is that the model is better fit without the addition of interaction term.

#### Sensitivity analyses

In table 2, the analysis was repeated by using baseline hs-CRP as a continuous independent variable in conducting the logistic regression. Due to its skewed distribution, the baseline hs-CRP in mg/L was first log transformed before assessing again its association with diabetes incidence at Tromsø7. Logistic regression, with or without multivariable adjusted for a priori covariates as described above, was used to evaluate the odds of diabetes cases by one-log increment of baseline hs-CRP among all study sample.

As discussed in the Methods section, individuals with baseline hs-CRP≥10 mg/L might be potentially experiencing acute inflammatory conditions at the time of the lab-based examination during the Tromsø Study survey.[53–58] To assess the robustness of the analysis, the

Table 1  Baseline characteristics of the study participants overall and across hs-CRP tertiles in the Tromsø Study 2007–2008

| | Tertiles of hs-CRP (mg/L) | | | | |
| | Tertile 1 (0.14–0.77) | Tertile 2 (0.78–1.69) | Tertile 3 (1.70–9.98) | P value* | Total |
|---|---|---|---|---|---|
| N | 2716 | 2673 | 2678 | | 8067 |
| Sex, n (%) | | | | | |
| Male | 1139 (41.9) | 1336 (50.0) | 1229 (45.9) | <0.001 | 3704 (45.9) |
| Female | 1577 (58.1) | 1337 (50.0) | 1449 (54.1) | | 4363 (54.1) |
| Age, n (%) | | | | | |
| 30–39 years | 108 (4.0) | 94 (3.5) | 78 (2.9) | <0.001 | 280 (3.5) |
| 40–44 years | 728 (26.8) | 485 (18.1) | 435 (16.2) | | 1648 (20.4) |
| 45–49 years | 332 (12.2) | 315 (11.8) | 267 (10.0) | | 914 (11.3) |
| 50–54 years | 325 (12.0) | 285 (10.7) | 274 (10.2) | | 884 (11.0) |
| 55–59 years | 268 (9.9) | 309 (11.6) | 301 (11.2) | | 878 (10.9) |
| 60–64 years | 471 (17.3) | 551 (20.6) | 598 (22.3) | | 1620 (20.1) |
| 65–69 years | 283 (10.4) | 349 (13.1) | 367 (13.7) | | 999 (12.4) |
| 70–74 years | 138 (5.1) | 184 (6.9) | 210 (7.8) | | 532 (6.6) |
| 75–87 years | 63 (2.3) | 101 (3.8) | 148 (5.5) | | 312 (3.9) |
| hs-CRP, median, mg/L | 0.50 | 1.12 | 2.83 | 0.0001 | 1.12 |
| (25th, 75th percentiles) | (0.38, 0.62) | (0.93, 1.39) | (2.13, 4.27) | | (0.62, 2.13) |
| BMI category, n (%) | | | | | |
| < 25 kg/m$^2$ | 1507 (55.5) | 865 (32.4) | 564 (21.1) | <0.001 | 2936 (36.4) |
| 25–29.9 kg/m$^2$ | 1048 (38.6) | 1362 (51.0) | 1270 (47.4) | | 3680 (45.6) |
| ≥ 30 kg/m$^2$ | 160 (5.9) | 445 (16.7) | 843 (31.5) | | 1448 (18.0) |
| Missing | 1 | 1 | 1 | | 3 |
| Hypertension, n (%) | 845 (31.2) | 1199 (45.0) | 1458 (54.5) | <0.001 | 3502 (43.5) |
| Missing | 5 | 6 | 4 | | 15 |
| Blood pressure, mean±SD | | | | | |
| Systolic, mm Hg | 127.65±20.35 | 134.22±22.11 | 137.52±21.54 | <0.0001 | 133.10±21.73 |
| Diastolic, mm Hg | 75.49±10.12 | 78.10±10.39 | 79.20±10.65 | <0.0001 | 77.59±10.50 |
| Blood pressure lowering drugs | | | | | |
| In use, n (%) | 355 (13.1) | 528 (19.8) | 717 (26.8) | <0.001 | 1600 (19.8) |
| Waist-to-height ratio | | | | | |
| Mean±SD | 0.52±0.06 | 0.55±0.06 | 0.58±0.07 | <0.0001 | 0.55±0.07 |
| Abdominal obesity, n (%) | 1646 (62.8) | 2125 (81.7) | 2341 (90.1) | <0.001 | 6112 (78.2) |
| Missing | 96 | 73 | 80 | | 249 |
| Non-HDL cholesterol, mmol/L | | | | | |
| Mean±SD | 3.85±1.02 | 4.20±1.06 | 4.32±1.08 | <0.0001 | 4.12±1.07 |
| Lipid-lowering drugs | | | | | |
| In use, n (%) | 260 (9.6) | 296 (11.1) | 315 (11.8) | 0.030 | 871 (10.8) |
| Alcohol consumption frequency, n (%) | | | | | |
| Never drinker | 172 (6.4) | 218 (8.2) | 242 (9.1) | <0.001 | 632 (7.9) |
| Monthly or less frequent | 617 (22.9) | 701 (26.5) | 799 (30.1) | | 2117 (26.5) |
| 2–4 times a month | 1146 (42.5) | 1088 (41.1) | 1054 (39.7) | | 3288 (41.1) |
| 2–3 times a week | 596 (22.1) | 501 (18.9) | 424 (16.0) | | 1521 (19.0) |

**Table 1** Continued

| | Tertiles of hs-CRP (mg/L) | | | | |
| | Tertile 1 (0.14–0.77) | Tertile 2 (0.78–1.69) | Tertile 3 (1.70–9.98) | P value* | Total |
|---|---|---|---|---|---|
| ≥4 times a week | 167 (6.2) | 141 (5.3) | 133 (5.0) | | 441 (5.5) |
| Missing | 18 | 24 | 26 | | 68 |
| Daily smoking, n (%) | | | | | |
| Current | 387 (14.4) | 469 (17.7) | 576 (21.8) | <0.001 | 1432 (17.9) |
| Previously | 1147 (42.6) | 1115 (42.1) | 1171 (44.3) | | 3433 (43.0) |
| Never | 1156 (43.0) | 1066 (40.2) | 899 (34.0) | | 3121 (39.1) |
| Missing | 26 | 23 | 32 | | 81 |
| Physical activity, n (%) | | | | | |
| Sedentary | 352 (13.6) | 383 (15.3) | 536 (21.6) | <0.001 | 1271 (16.8) |
| Low | 1580 (60.8) | 1532 (61.0) | 1542 (62.2) | | 4654 (61.3) |
| Moderate | 589 (22.7) | 552 (22.0) | 378 (15.2) | | 1519 (20.0) |
| Vigorous | 76 (2.9) | 44 (1.8) | 25 (1.0) | | 145 (1.9) |
| Missing | 119 | 162 | 197 | | 478 |
| Education, n (%) | | | | | |
| Primary/secondary | 516 (19.2) | 641 (24.2) | 762 (28.8) | <0.001 | 1919 (24.0) |
| 1–2 years senior high school | 618 (22.9) | 722 (27.3) | 746 (28.2) | | 2086 (26.1) |
| High school diploma | 220 (8.2) | 206 (7.8) | 212 (8.0) | | 638 (8.0) |
| College/university <4 years | 542 (20.1) | 504 (19.0) | 479 (18.1) | | 1525 (19.1) |
| College/university ≥4 years | 799 (29.7) | 575 (21.7) | 450 (17.0) | | 1824 (22.8) |
| Missing | 21 | 25 | 29 | | 75 |

*P value across hs-CRP tertiles was obtained by Pearson's $\chi^2$ test for categorial variables, by Kruskal-Wallis test for continuous variables with non-normal distribution, or by one-way ANOVA for continuous variables with normal distribution. ANOVA, analysis of variance; BMI, body mass index; HDL, high-density lipoprotein; hs-CRP, high-sensitivity C-reactive protein.

logistic regression models with or without multivariable adjusted for a priori covariates as described above were repeated by including study participants with hs-CRP≥10 mg/L using baseline hs-CRP as three quantiles and also as continuous independent variables (table 3).

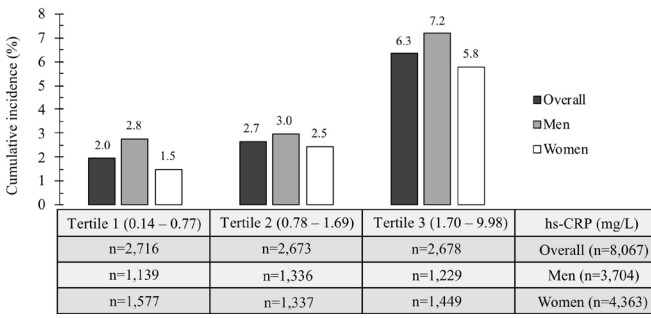

**Figure 3** Age-standardised cumulative diabetes incidence across hs-CRP tertiles during follow-up in the Tromsø Study 2015–2016. hs-CRP, high-sensitivity C-reactive protein.

All analyses were performed by using STATA V.14.2 (StataCorp. 2015. Stata Statistical Software: Release 14., StataCorp).

### Patient and public involvement

There was no patient involvement in the study. The original cohort data of the two surveys were collected with ethic approvals as described under ethics approval section in this paper. Only anonymous data from participants who had consented for medical research in the original Tromsø Study were used in this study. The participants from the original Tromsø Study did not participate in the development of the research question of this study nor the outcome measures of this study informed by patients' priorities, experience and preferences.

### RESULTS

### Baseline characteristics

As depicted in table 1, the total study sample of 8067 individuals had a median hs-CRP level of 1.12 mg/L. The median hs-CRP levels were 0.5 mg/L, 1.12 mg/L

**Table 2** Crude and multivariable adjusted logistic regression models for the association of hs-CRP and diabetes incidences in the Tromsø Study 2007–2016

| | Tertiles of hs-CRP (mg/L) | | | P value | Total |
|---|---|---|---|---|---|
| | Tertile 1 (0.14–0.77) | Tertile 2 (0.78–1.69) | Tertile 3 (1.70–9.98) | | |
| N | 2716 | 2673 | 2678 | | 8067 |
| Diabetes incidence*, n (%) | 52 (1.9) | 97 (3.6) | 171 (6.4) | <0.001† | 320 (4.0) |
| Logistic regression, hs-CRP in tertiles, OR (p value, 95% CI) | | | | | |
| Crude model | 1.00 (ref) | 1.93 | 3.49 | | |
| | | (<0.001, 1.37 to 2.71) | (<0.001, 2.55 to 4.79) | | |
| Multivariable adjusted‡ | 1.00 (ref) | 1.21 | 1.73 | | |
| | | (0.326, 0.83 to 1.78) | (0.004, 1.20 to 2.49) | | |
| Multivariable adjusted model+interaction term§ | p value, LRT | | | | |
| +hs-CRP tertile×sex | 0.902 | | | | |
| +hs-CRP tertile×BMI | 0.807 | | | | |
| +hs-CRP tertile×hypertension | 0.773 | | | | |
| +hs-CRP tertile×abdominal obesity | 0.632 | | | | |
| Logistic regression, hs-CRP as continuous variable, OR (p value, 95% CI) by one log increment of hs-CRP | | | | | |
| Crude model | 1.75 (<0.001, 1.53 to 1.99) | | | | |
| Multivariable adjusted‡ | 1.28 (0.003, 1.09 to 1.50) | | | | |

\* Known and unknown diabetes incidence in Tromsø7.
†p-value across hs-CRP tertiles by Pearson's $\chi^2$ test.
‡Adjusted for sex, age, body mass index, abdominal obesity, non-HDL cholesterol, lipid lowering drug use, hypertension, alcohol consumption frequency, daily smoking status, physical activity, and education.
§Models with hs-CRP in teriles adjusted for multivariables as described in ‡, except for variables included as interaction terms.
BMI, body mass index; HDL, high-density lipoprotein; hs-CRP, high-sensitivity C-reactive protein; LRT, likelihood ratio test; OR, odds ratio; ref, reference.

and 2.83 mg/L in tertile 1, 2, 3, respectively. Similarly, a previous study employing a large Norwegian community-based surveys, the Nord-Trøndelag Heath Study, reported a median hs-CRP level of 1.1 mg/L among the entire study sample of 30 669 aged from 20 and older.[65]

The baseline characteristics (waist-to-height ratio, abdominal obesity, non-HDL cholesterol, systolic/diastolic blood pressure, hypertension, blood pressure lowering drugs, daily smoking, alcohol consumption frequency, BMI, physical activity and education) of the

**Table 3** Sensitivity analysis including study participants with hs-CRP≥10 mg/L in the Tromsø Study 2007–2016

| | Tertiles of hs-CRP (mg/L) | | | P value | Total |
|---|---|---|---|---|---|
| | Tertile 1 (0.14–0.79) | Tertile 2 (0.80–1.78) | Tertile 3 (1.79–136.60) | | |
| N | 2801 | 2748 | 2754 | | 8303 |
| Diabetes incidence*, n (%) | 56 (2.0) | 101 (3.7) | 187 (6.8) | <0.001† | 344 (4.1) |
| Logistic regression, hs-CRP in tertiles, OR (p value, 95% CI) | | | | | |
| Crude model | 1.00 (ref) | 1.87 (<0.001, 1.34 to 2.60) | 3.57 (<0.001, 2.64 to 4.84) | | |
| Multivariable adjusted‡ | 1.00 (ref) | 1.13 (0.525, 0.78 to 1.64) | 1.68 (0.004, 1.18 to 2.40) | | |
| Logistic regression, hs-CRP as continuous variable, OR (p value, 95% CI) by one log increment of hs-CRP | | | | | |
| Crude model | 1.63 (<0.001, 1.47 to 1.81) | | | | |
| Multivariable adjusted‡ | 1.28 (<0.001, 1.12 to 1.46) | | | | |

*Known and unknown diabetes incidence in Tromsø7
†P value across hs-CRP tertiles by Pearson's $\chi^2$ test.
‡Adjusted for sex, age, body mass index, abdominal obesity, non-HDL cholesterol, lipid-lowering drug use, hypertension, alcohol consumption frequency, daily smoking status, physical activity and education.
HDL, high-density lipoprotein; hs-CRP, high-sensitivity C-reactive protein; OR, odds ratio; ref, reference.

participants exhibited statistical evidence for a difference across the three tertiles with increasing values of hs-CRP (p values <0.001 to <0.0001) (table 1). Conversely, lipid-lowering drug usage only showed modest impact (p=0.03). In tertiles 2 and 3, 55% and 60.5% of the participants were aged ≥55 years, respectively, while 82% and 90% of these participants were categorised as abdominal obesity. In addition, participants in tertiles 2 and 3 had higher percentage of overweight or obesity, hypertension, physically inactive, daily smokers, lower education and never or infrequent drinking when compared with those in tertile 1.

### Age-standardised cumulative diabetes incidence

There was a more than threefold increase of overall age-standardised cumulative diabetes incidence in the highest hs-CRP tertile 3 (6.3%) when compared with the reference tertile 1 (2.0%). This trend was apparent in both women (1.5% vs 5.8%, 3.9-fold increase) and men (2.8% vs 7.2%, 2.6-fold increase) (figure 3).

### Baseline hs-CRP associates with future diabetes development

There were 320 (4.0%) known and unknown incident cases of diabetes during follow-up (table 2). The odds of diabetes incidence were almost two times (crude OR 1.93; p<0.001; 95% CI 1.37 to 2.71) and 3.5 times (crude OR 3.49; p<0.001; 95% CI 2.55 to 4.79) higher among participants of hs-CRP tertile 2 and tertile 3, respectively. Although the strength of the association reduced in the fully adjusted model, positive association remained and the participants in hs-CRP tertile 3 had 73% higher odds of future diabetes (OR 1.73; p=0.004; 95% CI 1.20 to 2.49). The addition of the individual interaction term of hs-CRP tertile×sex (LRT p=0.902), hs-CRP tertile x BMI (LRT p=0.807), hs-CRP tertile×hypertension (LRT p=0.773) or hs-CRP tertile×abdominal obesity (LRT p=0.632) to the multivariable adjusted model did not support any compound effects of these covariates (table 2).

### Sensitivity analyses

In the sensitivity analysis depicted in table 2, consistent with the analyses using hs-CRP tertiles, a positive association between log transformed hs-CRP and diabetes development was observed. For every one-log baseline hs-CRP increment, there was 75% increase in odds of diabetes cases (crude OR 1.75; p<0.001; 95% CI 1.53 to 1.99). Positive association maintained after multivariable adjustment with 28% higher odds of diabetes development (OR 1.28; p=0.003; 95% CI 1.09 to 1.50) per one-log increase of baseline hs-CRP.

Another sensitivity analysis including study participants with hs-CRP≥10 mg/L was as described in table 3. There were additional 236 study participants with valid measurements included in this sensitivity analysis. With the exception of the upper limit of hs-CRP level in hs-CRP tertile 3, there was less than 0.1 mg/L in the changes of hs-CRP measurement among the three hs-CRP quantiles. The upper limit of hs-CRP tertile 3 had drastically

increased more than 13-fold to 136.60 mg/L, broadening particularly the range of hs-CRP measurement in tertile 3. There were 344 new known and unknown diabetes cases, accounting for 4.1% of these 8303 study samples, which is approximately 0.1% increase when comparing with the study samples excluding participants with hs-CRP≥10 mg/L in table 2. Even after including study participants with hs-CRP≥10 mg/L, consistent positive association was observed with 68% increase in multivariable adjusted odds (OR 1.68; p=0.004; 95% CI 1.18 to 2.40) of diabetes cases among the participants in hs-CRP tertile 3 (table 3). In general, by including participants with potentially acute inflammation (hs-CRP≥10 mg/L) conditions, the crude and adjusted associations were slightly biased towards null (table 2 vs table 3). Nevertheless, the models using hs-CRP as three quantiles remained robust in both scenarios. Similarly, when the sensitivity analysis was repeated among these 8303 study samples using hs-CRP as continuous independent variable, the crude OR also slightly biased towards null (table 2 vs table 3), denoting a 63% increase in crude odds of new diabetes cases (crude OR 1.63; p<0.001; 95% CI 1.47 to 1.81) by one log increment of hs-CRP. Nevertheless, the multivariable adjusted association (OR 1.28; p<0.001; 95% CI 1.12 to 1.46; table 3) per one-log increase of baseline hs-CRP remained robust even after including participants with potentially acute inflammation (hs-CRP≥10 mg/L) conditions.

## DISCUSSION

We found a marked association between high hs-CRP and future diabetes incidence at 7-year follow-up in this study of adults from a general Norwegian population.

### Comparison to other studies

A prospective study conducted in Spain demonstrated a positive association between elevated hs-CRP with future diabetes development after adjusting for age, sex and obesity but a study in UK found that the association was attenuated and no longer significant after adjusting for BMI and waist-to-hip ratio.[36 39] While some others showed that the association became insignificant in men but maintained in women after adjusting for BMI, suggesting evidence of interaction between hs-CRP and sex on the association.[34 40] However, this study did not observe evidence for interaction between hs-CRP and sex, BMI, abdominal obesity or hypertension. The association was also maintained after adjusting for multiple factors including BMI and abdominal obesity. Whereas some other studies did not exclude individuals with elevated baseline CRP levels above 10 mg/L,[34 36] which might have uplifted the exposure level due to the potential underlying issues of acute inflammatory condition other than the risk associated with diabetes development. This heightened exposure threshold likely attenuated the true association of the exposure and outcome by means of non-differential misclassification. Indeed, a slight

bias towards null was observed in our sensitivity analyses (table 2 vs table 3) after including study participants with hs-CRP≥10 mg/L, especially when stratifying participants into three hs-CRP quantiles. Nevertheless, the positive association remained robust in both scenarios.

A Finnish cohort study relied primarily on the diagnosed diabetes data through medical registries and self-reported surveys,[40] whereas a German case–control study identified baseline diabetes solely by self-reported information.[34] There might be limitation of misclassifying baseline diabetes due to bias on self-reported diabetes. The lack of validated laboratory-based tests such as HbA1c levels or fasting-blood sugar content in these studies might fail to capture any unknown diabetes at baseline, which might have led to non-differential misclassification and could have diluted the strength of the association. Whereas, in the German study, the differential misclassification of the baseline unknown diabetes among the controls might have biased the association leaning towards null due to the case–control study design,[34] which in part might be explaining the attenuation in the strength of association after adjusting for BMI. Lee et al,[39] on the other hand, used a HbA1c level cut-off of >7.0% (ie, >53 mmol/mol) for identifying unknown diabetes status at baseline and during follow-up. While this higher cut-off had been used for defining diabetes status, this deviation from the cut-off of 6.5% (ie, 48 mmol/mol) recommended by the WHO could introduce inconsistencies with the findings of other studies which chose to follow the WHO recommended standards.[66]

There is increasing awareness that elevated hs-CRP levels and diabetes incidences are associated with adiposity.[26] The association of elevated hs-CRP with central adiposity could be independent of BMI.[67] Some European studies adjusted their models for BMI but not for central adiposity.[34 36 40] Waist-to-hip ratio has been indicated using various sex-specific thresholds.[68] Although Lee et al had adjusted for both BMI and waist-to-hip ratio,[39] the inherent limitation of the waist-to-hip ratio might have led to residual confounding in some of these published analyses. Indeed, Lee et al suggested that the differential results found in a meta-analysis of 16 published studies might be due to, in part, the adjustment for different indexes of central adiposity.[39] The waist-to-height ratio had been found to be sex-independent, age-independent and ethnicity-independent[68]; and the use of this index in the current study might have addressed more adequately the confounding impact of the fat storage profile for both men and women.

Aging-associated chronic inflammation has been proposed as playing a part in the aetiology of type 2 diabetes.[69] Although ageing has long been identified as a risk factor and positively correlates with pathogenesis, it would be challenging to pinpoint a particular age cut-off to inform clinical practice. Previous research have suggested multiple sources of aging-associated chronic inflammation including age-related accumulation of damaged cells, changes in gut microbiota over

age, increasing cellular senescence and the subsequent secretion of proinflammatory cytokines namely the senescence-associated secretory phenotype or SASP, and age-associated modification of the immune system among others.[69] Although fat is a good source of inflammatory cytokines, obesity alone may not represent the entire spectrum of mechanisms underlying the development of diabetes via chronic systemic inflammation.

Individuals deemed pre-diabetes, defined by HbA1c between 5.7% and 6.4% or 39–47 mmol/mL, were included in this current study. Pre-diabetes state has been considered as risk for diabetes transformation.[70] Although positive association had been previously implicated,[71 72] the temporal relationship between elevated hs-CRP and pre-diabetes condition remained unclear. Glycaemic parameters such as HbA1c could provide estimates to reflect the average blood glucose concentration of an individual over the preceding 2–3 months. Nevertheless, the method also elicits certain extent of inaccuracies due to some glucose-independent variation, although corrective modelling has recently been proposed.[73]

### Strengths of the study

This study has several strengths. The 7-year follow-up study allowed us to investigate the temporal association of hs-CRP on diabetes development. The study population was recruited and sampled from all residents of the Tromsø municipality with validated systematic data collection for each of the repetitive surveys, including the questionnaires, the clinical examinations and the lab assessments. The data being analysed were of high quality with high participation in the study, and low levels of missing data in general (<11%)—which garners high internal validity. No findings from similar population-based studies conducted in the Scandinavian countries investigating the association between CRP and incident diabetes have been previously reported. The current findings of the positive CRP–diabetes association may inform public health strategies for prevention and surveillance of diabetes in the Nordic region where lifestyle, culture, demography, healthcare and socioeconomic factors resemble Norway.

The data available in the current analysis included the information of the use of lipid-lowering drugs, which are used in primary and secondary prevention of cardiovascular diseases but have previously been shown capable in reducing hs-CRP levels but at the same time might contribute to drug-induced side effects of impaired glycaemic control or new onset of diabetes.[31] These features might have led to residual confounding in some other previous studies which were not able to or did not include medication use in their multivariable adjusted models.[15 33] Moreover, the current study had used a large overall cohort sample size.

### Limitations of the study

Limitations of this study include the lack of information of diabetes subtype. Since the study sample only included

adults 30 years of age and above, primarily newly developed type 2 diabetes were expected. The study focused on a Norwegian general population and hence the conclusion may not be generalisable beyond this region. Since the baseline and future developed diabetes cases were self-reported, there might be potential non-differential misclassification of the outcome due to reporting bias. Similarly, self-reported medication for treating symptoms did not come with physician, medical record or medical registry confirmation might have led to potential measurement error in the outcome. Nevertheless, previous studies have found that self-reported diabetes often accounted for high specificity (84.5%–99.7%) although it can exhibit low sensitivity (41.5%–80.4%).[74–76] Previous studies had reported the impact from socioeconomic position and demographic characteristics such that the sensitivity of self-reported diabetes can be elevated with higher education levels and among urban dwellers.[76] In the current study, validated laboratory-based measurement of the HbA1c level was used in addition to self-reported diabetes and diabetes medication use for treating symptoms to enhance both the sensitivity and specificity when capturing the known and unknown true positive and false negative diabetes cases.

We did not include information on diet, which is another potential confounder. Diet can be negatively or positively associated with hs-CRP level and with future diabetes development depending on the diet types and patterns.[77] It would be of interest to include this parameter in future studies. In addition, diet is also linked with socioeconomic status which was apparently correlated with hs-CRP level as depicted in this report. On the other hand, due to data sparsity, we had combined participants with underweight condition ($<18.5\,\mathrm{kg/m^2}$) with those with normal BMI 18.5–24.9 $\mathrm{kg/m^2}$) as a single entity BMI ($<25\,\mathrm{kg/m^2}$) in the model. Given that the overweight and obesity conditions were more likely associated with hs-CRP and/or diabetes development,[12 25–27] the grouping of the small number of underweight with the normal BMI participants should not introduce impactful confounding effects in the model.[78]

As chronic inflammation relates to the aetiology of type 1 diabetes, it would be of interest to study its association with hs-CRP by utilising a more defined type 1 diabetes cohort. It is also of interest to include hormone replacement therapy as a potential confounder since this medication for treating symptoms had been implicated associated with elevated hs-CRP levels but also implicated in the reduction of abdominal fat.[79 80]

## CONCLUSION

In conclusion, this study has demonstrated that hs-CRP is associated with future diabetes development in a population-based sample of Norwegian adults. This association was consistent across sex, BMI, abdominal obesity and hypertension. In combination with glycaemic parameters such as HbA1c or fasting plasma glucose, this stable blood-based biomarker can potentially be used to facilitate risk group stratification, which may provide opportunities for early prevention or intervention of diabetes through clinical practices, public health education and anti-inflammatory therapeutics development. Previous studies had depicted ethnic variations of CRP levels which we were not able to explore within this study.[80 81] When comparing with white, after multivariable adjustment including BMI, black individuals had higher serum CRP concentrations while Asians including Chinese and Japanese had lower levels. Future studies to determine the potentially ethnicity-dependent cutoffs of hs-CRP for risk stratification are warranted.

**Acknowledgements** We would like to thank the participants of the Tromsø Study. We would like to extend our gratitude to Dr. Anne Elise Eggen, Dr. Ola Løvsletten, Ms. Julie-Helene D. Sørensen, and the members of the Tromsø Study Data and Publication Committee of UiT The Arctic University of Norway to make the Tromsø dataset available for this study.

**Contributors** KIT conceived the research question of the present study based on the suggestion from SC on a project relating to diabetes and the Tromsø Study. KIT planned out the study design, analysed and interpreted the data, constructed figures and tables, wrote the manuscript. LAH and SC contributed to the discussion of the research question, study design, analysis and data interpretation. LAH and SC took part in the critical revision of the manuscript and provided overall supervision of the study. SC coordinated the collaboration of the study. LAH took part in the data collection of Tromsø6 and Tromsø7. LAH took part in the planning and management of Tromsø7. KIT is the guarantor of this work and, as such, had full access to all the data in the study and takes responsibility for the integrity of the data and the accuracy of the data analysis. All authors approved the final manuscript.

**Funding** Funding for the Tromsø Study was obtained from UiT The Arctic University of Norway, Northern Norway Regional Health Authority, Ministry of Health and Care Services, Norwegian Research Council, and various public and charity research funds in Norway. For SC infrastructure support for this research was provided by the NIHR Imperial Biomedical Research Centre.

**Disclaimer** There was no external funding for this specific paper. The funding bodies have no role in the design of the study, data collection, analysis, interpretation of data, or in writing the manuscript.

**Competing interests** None declared.

**Patient and public involvement** Patients and/or the public were not involved in the design, or conduct, or reporting, or dissemination plans of this research.

**Patient consent for publication** Not applicable.

**Ethics approval** This study involves human participants and Tromsø6 was approved by the Data Inspectorate of Norway and the Regional Committee for Medical and Health Research Ethics Northern Norway (REC North) (Ref. 121/2006) and Tromsø7 by the Norwegian Data Protection Authority (Ref. 14/01463-4/CGN) and REC North (Ref. 2014/940). The current study was approved by the London School of Hygiene & Tropical Medicine Ethics Committee (Ref. 2566). Participants gave informed consent to participate in the study before taking part.

**Provenance and peer review** Not commissioned; externally peer reviewed.

**Data availability statement** Data may be obtained from a third party and are not publicly available. Data are not publicly available but may be obtained from UiT The Arctic University of Norway through application and project approval by the Data and Publication Committee. The data used in this study are owned and managed by UiT The Arctic University of Norway (https://uit.no/research/tromsostudy).

**ORCID iD**
Sarah Cook http://orcid.org/0000-0003-1250-2967

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
