## [Reviewer comments · BMJ Open]

ARTICLE DETAILS

TITLE (PROVISIONAL)	The association of C-reactive protein with future development of diabetes: A population-based 7-year cohort study among Norwegian adults aged 30 and older in the Tromsø Study 2007 – 2016
AUTHORS	Tong, Kit I.; Hopstock, Laila; Cook, Sarah

VERSION 1 – REVIEW

REVIEWER	Naheed, Aliya International Centre for Diarrhoeal Disease Research Bangladesh, Initiative for Noncommunicable Diseases
REVIEW RETURNED	25-Jan-2023

GENERAL COMMENTS	General appraisal The objective of the study was to investigate the predictive value of hs-CRP for diabetes development in a Norwegian general. The area of research is important and the study has found that there is a positive association between the hs-CRP and incident of diabetes. However, the authors did not present any conceptual framework or rationale for choosing the variables included in the analytical framework for investigating the role of hs-CRP in developing diabetes through analyses of multi-year cohort data. The methodology adopted in the research is not robust to answer the research questions framed by the authors. The authors should develop a predictive model before presenting multi stage analyses. The findings are insufficient to support the case for Norwegian adults and the region as a whole. The article is also not developed following the standard guideline of presenting manuscript that would include a discussion followed by a conclusion. The authors may follow the comments below for improving the quality of the manuscript. Major comments: Abstract It is unclear whether self-reported diabetes was verified with prescription or a lab investigation. The WHO definition of diabetes does not include self-report for diagnosis of diabetes. Some demographic characteristics must be described, preferably in a single sentence. Key message of table 1 can be highlighted in the abstract. It would be preferable if the author could report incidence rate rather than incidence proportion. When reporting OR and 95% CI, the author can ignore the P-value.
--

	Background and rationale of the study Author can report the updated estimate of global burden of diabetes from the GBD studies. For example, PouyaSaeedi et al. "Global and regional diabetes prevalence estimates for 2019 and projections for 2030 and 2045: Results from the International Diabetes Federation Diabetes ; At least, 9th edition; Diabetes Research and Clinical Practice". The study's rationale is not well explained. The authors should describe the general state of metabolic risk in Norway. It would be useful to report the average hs-CRP concentration in Norway, as well as why such investigation is important for this country and the region as a whole. The relationship between the elevation of C-reactive protein (CRP) and development of type 1 and type 2 diabetes is supported by a single reference, which is not strong enough to justify the rationale of the study. The rationale of excluding individuals with hs-CRP level $\geq 10\text{mg/L}$ indicating an acute inflammatory condition was not understood, since an acute exacerbation of a chronic inflammation might be an important predictor for diabetes. The rationale of the study should be well supported by robust scientific evidence. Case definition and study population Diabetes was defined by self reporting or use of medication or insulin or using HbA1C cut off. None of those parameters are used for diagnosis of a new case of diabetes and HbA1C is more used for a clinical assessment of control of diabetes. Since the objective of the study is to explore chronic inflammation as a predictor of diabetes incidence (new case) more robust clinical evidence, such as laboratory confirmed diabetes would be a desired for a case definition for diabetes, such as OGTT. Selection of the study samples from different study cohort was not clearly stated. The authors should justify how the selection process across different cohorts was standardized and there was no potential selection bias. The justification of using different parameters in baseline and follow up for exclusion is not clear. However, all of the parameters used for exclusion did not confer diagnosis of diabetes. Analytical plan The analytical plan is not supported by any hypothesis and biological plausibility of different biomedical parameters attributing to development of a new case of diabetes due to chronic inflammation is not supported by existing evidence while choosing the analytical tools. The selection of so many variables in the multivariable model and introducing interaction terms are not justified. A brief discussion about the variables and source of the selected variables should be reported. The recruitment strategy could be more specific about how many people were selected in Troms 6 and Troms 7. The variable selection strategy can be more specific with rationale for the multivariable model. Why interaction analysis is required for this study should be explained in the rationale section. What strategy did the authors use to define the confounding and interaction variables that needed to be discussed in the data analysis plan.. The interaction analysis
--	---

	can be detailed and presented in a separate tables. The key outcome should be confirmed by sensitivity analysis to explore whether the association between hs-CRP and incident of diabetes exist in a sub-group analysis in order to support discussion. In this study how the level of statistical significance was determined is unclear? Author should mention the cutoff of the probability value for the level of significance. How physical activity status was assessed in the study is unclear. It is preferable to use a globally accepted tool to measure physical activity rather than self-reported data. In this study, author have used BMI (< 25 kg/m²) for normal weight range which is not correct. The WHO criteria or any others globally acceptable cut off values can be adopted for the analysis. The methodology lacks an appropriate reference for each cut off value for anthropometric and clinical assessment. Why the hs-CRP level stratified into three categories should be justified and if there is any clinical implication of presenting stratified analysis for three tertile should be justified. The authors have stratified the hs-CRP data into three different tertial which transformed a continuous data into ordinal data. hs-CRP level were log transformed due to non-normal structure. It is not mentioned why the authors did not follow a similar strategy for others variables (blood glucose, HDL cholesterol). Data interpretation and conclusion Description of basic characteristics is unclear in the result section. A baseline difference between patients with and without diabetes is required to understand how the distribution of study participants differs according to the disease status. Age group has not been categorized in the consistent manner. In Table 1, the author should specify the type of statistical test applied to asses changes of mean and SD across tertile of hs-CRP level. P-value for trend can be appropriate in this case to report mean or median changes across tertiles of hs-CRP level. In Table 2, the author used a logistic regression model to demonstrate the relationship between hs-CRP and diabetes incidence. It was unclear how the model was adjusted for the correlated structure of data (repeated measure of same individuals) and how the time difference was adjusted. The author can report an incidence rate ratio instead of odds ratio, which may provide better epidemiological insight into this analysis. Conclusion Authors can interpret the findings in a separate chapter like discussion and conclusion. The primary goal of the study was to investigate the association between hs-CRP and diabetes. However, the current analysis is insufficient to justify whether the association is independent of others demographic, anthropometric and clinical variables. Authors should investigate the association between hs-CRP and diabetes incident in terms of sensitivity analysis.
--	--

	The authors can describe the findings contextualizing the key messages in terms of sociodemographic, clinical and anthropometric factors of diabetes patients. In final conclusion, hs-CRP has been identified as a predictor of future diabetes development. The current analysis, however, does not support this statement, and the findings only show a positive association that needs to be confirmed by further sub-analysis. The conclusion should be rewritten and revised in light of the study's objective and key messages. Minor comments:  1. Author can remove P-value if report Odds ratio and 95% CI of Odds Ratio. 2. Caption of Table 1 is missing. 3. In this analysis, cox regression can be applied by sequential model to estimate hazard ratio for the incident of diabetes after adjusted with different factors (clinical, anthropometric, medications). 4. Please move this sentence into the data analysis part "the distribution of CRP was skewed and natural log-transform was used for the statistical analysis". 5. Please use the term "Multivariable" instead of "Multivariate" as because multivariable indicates an association between single response/outcome variable and multiple explanatory variables.
--	---

REVIEWER	Akinboboye, Olaitan Medical College of Wisconsin, Institute of Health and Equity
REVIEW RETURNED	17-Feb-2023

GENERAL COMMENTS	The article's title is "The predictive value of C-reactive protein for future development of diabetes in a general population: The Tromsø Study 2007-2016." This study aimed to determine the predictive value of hs-CRP for diabetes development in a Norwegian general population. It sounds interesting, primarily because of the sample population. However, a few concerns need to be addressed.  1. For the multiple regression analysis, please clarify the methods for variables selection into the model. 2. What informed the values (upper and lower limits) used in grouping the hs-CRP levels into tertiles? 3. How did the authors handle pre-diabetes individuals with HbA1c between 5.7% – 6.4%? 4. Does the Tromsø Study collect cancer information? If yes, did the authors consider controlling for cancer?
--

VERSION 1 – AUTHOR RESPONSE

Reviewer: 1

Dr. Aliya Naheed, International Centre for Diarrhoeal Disease Research Bangladesh

Comments to the Author:

General appraisal

The objective of the study was to investigate the predictive value of hs-CRP for diabetes development in a Norwegian general. The area of research is important and the study has found that there is a positive association between the hs-CRP and incident of diabetes. However, the authors did not present any conceptual framework or rationale for choosing the variables included in the analytical framework for investigating the role of hs-CRP in developing diabetes through analyses of multi-year cohort data. The methodology adopted in the research is not robust to answer the research questions framed by the authors. The authors should develop a predictive model before presenting multi stage analyses.

Authors' response:

We thank you Dr. Naheed for the insightful comment.

In response to the reviewer, we have added in the text: (1) In introduction, in paragraphs 4-6 on page 6-7, (2) in Methods under sections "Variables in the analyses" on pages 11-13, and "statistical analysis" on pages 13 and 18, (3) and also "Figure 1 - conceptual framework" to clearly describe the rationale for selecting the variables as potential confounders and/or potential effect modifiers in the model, the definition of the variables, and how the variables were used in the model analyses. In brief, potential confounders were those that have been indicated in previous studies associated with CRP levels and/or with diabetes incidences, but not mediating the pathway between the association of CRP and diabetes incidences. Effect modifiers were those that have been indicated interacting with hs-CRP but exhibiting conflicting conclusions in previous studies.

We have also rephrased our title from "predictive value" to investigate "the association" of hs-CRP for future diabetes development and objectives of the study - to better suit the entire study presented here.

The dataset involved 2 population-based surveys from the Tromsø Study. The two surveys were conducted separately with one in 2007-2008 that was defined as baseline and the other one in 2015-2016 defined as the follow-up time point after a 7-year interval. There was no data collection between the years of the two surveys.

The dataset of this study, therefore, is not permissible to conduct multi-stage follow-up analyses but rather is more suitable for one-time point follow-up analysis design in this study.

Therefore, we have edited the Methods under "Setting" on page 8 to clarify the nature of the dataset used in this study. We have also added "study design" in Methods to describe the overall strategy of the study on page 8.

The findings are insufficient to support the case for Norwegian adults and the region as a whole. The article is also not developed following the standard guideline of presenting manuscript that would include a discussion followed by a conclusion. The authors may follow the comments below for improving the quality of the manuscript.

Authors' response:

We thank you Dr. Naheed for the insightful comment.

Authors acknowledge the limitation of this study as it focused on a Norwegian general population and hence the conclusion may not be generalizable beyond this region.

We have, therefore, include this discussion on page 25 under "Limitations of the study".

We apologized for our typo error of the heading “CONCLUSION” in our first manuscript. We have corrected it to “DISCUSSION” for pages 22-26 and “CONCLUSION” for page 27 in the revised version.

Major comments:

Abstract

It is unclear whether self-reported diabetes was verified with prescription or a lab investigation. The WHO definition of diabetes does not include self-report for diagnosis of diabetes.

Authors’ response:

We thank you Dr. Naheed for the insightful comment.

Authors acknowledge the limitation of including the use of self-reporting diabetes and use of diabetes-related medications for treating symptoms. Our data is taken from population-based survey data, which does not include clinical health registries.

To identify diabetes cases, we used (1) self-reported diabetes, (2) self-reported medication for diabetes for treating symptoms, and also (3) a more robust lab-based blood measurement of glycated hemoglobin (HbA1c) to support both the data of self-reported diabetes and diabetes medication use for treating symptoms. By using the lab-based blood test of HbA1c levels of participants, we believe that would increase the sensitivity and specificity of the diabetes case definition.

Although it may not be a global practice, WHO recommends HbA1c diagnostic threshold of $\geq 6.5\%$ (≥ 48 mmol/mol) for diabetes in 2011 [1]

Other countries such as UK [2], USA [3] Canada [4] adapted WHO diagnostic guidelines of $\geq 6.5\%$ (≥ 48 mmol/mol) HbA1c for Type 2 diabetes in adults.

Norwegian Institute of Public Health [5] regards HbA1c, which can be measured without fasting blood sample, a good alternative to fasting blood glucose level. HbA1c is the current recommended diagnosis definition for diabetes in Norway with HbA1c ≥ 48 mmol/mol ($\geq 6.5\%$) as diagnostic threshold [5]. This recommendation corroborates with WHO’s guidance [1]

References:

[1] Use of glycated haemoglobin (HbA1c) in the diagnosis of diabetes mellitus. Abbreviated report of a WHO consultation. [Article online], 2011.

Available: https://apps.who.int/iris/bitstream/handle/10665/70523/WHO_NMH_CHP_CPM_11.1_eng.pdf?sequence=1&isAllowed=y

[2] <https://www.diabetes.org.uk/professionals/position-statements-reports/diagnosis-ongoing-management-monitoring/new-diagnostic-criteria-for-diabetes>

[3] <https://www.cdc.gov/diabetes/basics/getting-tested.html>

[4] https://www.diabetes.ca/health-care-providers/clinical-practice-guidelines/chapter-3#panel-tab_FullText

[5] <https://www.fhi.no/en/op/Indicators-for-NCD/premature-mortality/diabetes-hos-voksne-indikator-12/>

We have included discussion on pages 25-26 under “limitations of the study” regarding the limitations of using self-reported diabetes and self-reported diabetes medication use for treating symptoms. We have also included discussion on how the additional use of lab-based measurement of HbA1c would improve the sensitivity and specificity in identifying true diabetes cases among the study samples in the study.

Some demographic characteristics must be described, preferably in a single sentence. Key message of table 1 can be highlighted in the abstract.

Authors' response:

We thank Dr. Naheed for the insightful comment.

We'd added description, in the abstract on page 2, on demographic characteristics under "setting" and had also highlighted the key message of table 1 under "Results".

It would be preferable if the author could report incidence rate rather than incidence proportion.

Authors' response:

We thank Dr. Naheed for the insightful comment.

Since the data was taken from 2 population-based surveys of two discrete times - the baseline in Tromsø6 (2007-2008) and follow-up time in Tromsø7 (2015-2016) - therefore, we do not have the exact follow-up time of diabetes cases to report incidence rate as there was no data collection during the years between the 2 surveys.

In Methods under "Setting" on page 8, we'd added the background of the Tromsø Study and clarified the nature of the dataset with the use of Tromsø6 survey as baseline and Tromsø7 as the follow up time point after a leap of 7-year interval. We have also added "study design" in Methods to describe the overall strategy of the study.

When reporting OR and 95% CI, the author can ignore the P-value.

Authors' response:

We thank Dr. Naheed for the insightful comment.

Authors think that it would be more informative to include P-value when reporting OR and 95% CI in Table 1, since that would inform how different each baseline characteristics across the three hs-CRP levels among the study samples.

Background and rationale of the study

Author can report the updated estimate of global burden of diabetes from the GBD studies. For example, PouyaSaeedi et al. "Global and regional diabetes prevalence estimates for 2019 and projections for 2030 and 2045: Results from the International Diabetes Federation Diabetes; At least, 9th edition; Diabetes Research and Clinical Practice"

Authors' response:

We very much appreciate Dr. Naheed for this very insightful suggestion.

We have updated the 2019 global estimate of diabetes both in the text at the beginning of introduction on page 5, first paragraph, and also replaced reference 1 with the Saeedi P *et al.* paper.

The study's rationale is not well explained. The authors should describe the general state of metabolic risk in Norway.

Authors' response:

We thank Dr. Naheed for this insightful suggestion.

We have added the description in the introduction section, on page 7 second last paragraph, about the previous studies conducted in Northern Norway regarding the increasing prevalence of metabolic syndrome and severity in Northern Norway.

In brief:

Using the most recent revised consensus definition of metabolism syndrome from the harmonized Adult Treatment Panel-III (ATP-III) criteria [ref 48 in the manuscript], a recent repeated large cohort cross-sectional study of the rural Northern Norway has found that the prevalence of age-standardized metabolic syndrome (MetS) has progressed from 31.2% in 2003-2004 (n=6550) to 35.6% in 2012-2014 (n=6004) with concomitant increase in the mean MetS severity, which is believed to be driven by the marked increase in abdominal obesity among the study population over the years [ref 49 in the manuscript].

On page 7, we have also added description of information from Norwegian Institute of Public Health that there was high prevalence of overweight and obesity combined in Norway [ref 47 in the manuscript]

It would be useful to report the average hs-CRP concentration in Norway, as well as why such investigation is important for this country and the region as a whole.

Authors' response:

We thank Dr. Naheed for this insightful suggestion.

Authors do not aware of any national health data on average hs-CRP level in Norway. However, the dataset in this study showed a median of hs-CRP level of 1.12mg/L from 8,067 study participants of the population-based Tromsø Study survey from a general Norwegian population aged 30 to 87 years old.

Another study using a large Norwegian community-based Nord-Trøndelag Health Study (HUNT) surveys which involved with 30,669 individuals aged from 20 and older also reported a median hs-CRP level of 1.1 mg/L of the entire study sample. The observation of this previous study corroborates with our observation.

[ref 59 in the manuscript]

We have included this data in Table 1 and added the description in Results under "Baseline characteristics" on page 20, first paragraph.

The relationship between the elevation of C-reactive protein (CRP) and development of type 1 and type 2 diabetes is supported by a single reference which is not strong enough to justify the rationale of the study.

Authors' response:

We thank Dr. Naheed for the insightful comment.

In addition to the original ref 15, we'd expanded the introduction and increased the number of references related to the relationship of raised CRP and the development of type 1 and type 2 diabetes [ref 4,5,9-14,16 in the manuscript] on pages 5-6, paragraph 3 of Introduction.

In paragraph 5 of the Introduction section on page 6 we have further discussed the heterogenous conclusions of previous studies on the association between CRP and diabetes with references 15, 33-39 to further reinforce the rationale and the need of this current study.

The rationale of excluding individuals with hs-CRP level ≥ 10 mg/L indicating an acute inflammatory condition was not understood, since an acute exacerbation of a chronic inflammation might be an

important predictor for diabetes. The rationale of the study should be well supported by robust scientific evidence.

Authors' response:

We thank Dr. Naheed for the insightful comment.

Since acute inflammation may resolve after a short period of time and may not necessarily always transit subsequently into chronic inflammation, we have a concern that by including hs-CRP level $\geq 10\text{mg/L}$, an indication of acute inflammatory condition, may contribute to the risk of measurement error.

Indeed, a previous study among 5,111 men, aged 48-77 years, in the Oslo Study showed that men with conditions of osteoporosis, asthma, chronic bronchitis/emphysema, diabetes, or myocardial infarction had mean CRP values of 6.53mg/L, 5.01mg/L, 4.42mg/L, 4.53mg/L, and 4.27mg/L, respectively.

The mean CRP levels among men even with conditions such as diabetes, cardiovascular disease, and chronic inflammatory disorders were much less than 10mg/L.

We, therefore, believe that by excluding hs-CRP level $\geq 10\text{mg/L}$ may mitigate the risk of measurement error in the current study [ref 53 in the manuscript]

We have included this additional rationale in the Methods section under "Study sample" second paragraph on page 9.

Case definition and study population

Diabetes was defined by self reporting or use of medication or insulin or using HbA1C cut off. None of those parameters are used for diagnosis of a new case of diabetes and HbA1C is more used for a clinical assessment of control of diabetes. Since the objective of the study is to explore chronic inflammation as a predictor of diabetes incidence (new case) more robust clinical evidence, such as laboratory confirmed diabetes would be a desired for a case definition for diabetes, such as OGTT.

Authors' response:

We thank Dr. Naheed for the insightful comment.

Authors acknowledge the limitation of including the use of self-reporting diabetes and use of diabetes-related medications for treating symptoms. Our data is taken from population-based survey data, which does not include clinical health registries. The study aim was to examine the association between baseline hs-CRP and future diabetes development.

To identify diabetes cases, we used (1) self-reported diabetes, (2) self-reported medication for diabetes for treating symptoms, and also (3) a more robust lab-based blood measurement of glycated hemoglobin (HbA1c) to support both the data of self-reported diabetes and diabetes medication use for treating symptoms. By using the lab-based blood test of HbA1c levels of participants, we believe that would increase the sensitivity and specificity of the diabetes case definition.

Although it may not be a global practice, WHO recommends HbA1c diagnostic threshold of $\geq 6.5\%$ ($\geq 48 \text{ mmol/mol}$) for diabetes in 2011 [1]

Other countries such as UK [2], USA [3] Canada [4] adapted WHO diagnostic guidelines of $\geq 6.5\%$ ($\geq 48 \text{ mmol/mol}$) HbA1c for Type 2 diabetes in adults.

Norwegian Institute of Public Health [5] regards HbA1c, which can be measured without fasting blood sample, a good alternative to fasting blood glucose level. HbA1c is the current recommended

diagnosis definition for diabetes in Norway with HbA1c ≥ 48 mmol/mol ($\geq 6.5\%$) as diagnostic threshold [5]. This recommendation corroborates with WHO's guidance [1]

References:

[1] Use of glycated haemoglobin (HbA1c) in the diagnosis of diabetes mellitus. Abbreviated report of a WHO consultation. [Article online], 2011.

Available: https://apps.who.int/iris/bitstream/handle/10665/70523/WHO_NMH_CHP_CPM_11.1_eng.pdf?sequence=1&isAllowed=y

[2] https://www.diabetes.org.uk/professionals/position-statements-reports/diagnosis-ongoing-management-monitoring/new_diagnostic_criteria_for_diabetes

[3] <https://www.cdc.gov/diabetes/basics/getting-tested.html>

[4] https://www.diabetes.ca/health-care-providers/clinical-practice-guidelines/chapter-3#panel-tab_FullText

[5] <https://www.fhi.no/en/op/Indicators-for-NCD/premature-mortality/diabetes-hos-voksne-indikator-12/>

We have included discussion on pages 25-26 under "limitations of the study" regarding the limitations of using self-reported diabetes and self-reported diabetes medication use for treating symptoms. We have also included discussion on how the additional use of lab-based measurement of HbA1c would improve the sensitivity and specificity in identifying true diabetes cases among the study samples in the study.

Selection of the study samples from different study cohort was not clearly stated. The authors should justify how the selection process across different cohorts was standardized and there was no potential selection bias. The justification of using different parameters in baseline and follow up for exclusion is not clear. However, all of the parameters used for exclusion did not confer diagnosis of diabetes

Authors' response:

We thank Dr. Naheed for the insightful comment.

We have edited the text and included figure 2 to better describe the selection process of the study sample in Methods under "Study sample" on pages 9-10.

In brief, the study sample was selected from the sixth survey of the Tromsø Study (2007-2008) with the following criteria:

- (1) They are non-diabetes at baseline during the sixth survey of the Tromsø Study (2007-2008)
- (2) Baseline hs-CRP < 10 mg/L
- (3) The same participants had also attended the seventh survey of the Tromsø Study (2015-2016) with valid data to determine their diabetes status during the Tromsø7 survey.

Analytical plan

The analytical plan is not supported by any hypothesis and biological plausibility of different biomedical parameters attributing to development of a new case of diabetes due to chronic inflammation is not supported by existing evidence while choosing the analytical tools. The selection of so many variables in the multivariable model and introducing interaction terms are not justified. A brief discussion about the variables and source of the selected variables should be reported

Authors' response:

We thank Dr. Naheed for the insightful comment.

We have expanded the description on the hypothesis, the biological plausibility in the introduction on pages 5-7 to better support the rationale for the analysis.

We have added additional details and references in the introduction in paragraphs 4 & 6 on pages 6-7 regarding the selection of the potential confounders based on previous studies. In addition, we have added Figure 1 - the conceptual framework to illustrate the relationship of the selected variables (potential confounder or comorbidity) with hs-CRP (exposure) and diabetes (outcome).

In brief, potential confounders were those that have been indicated in previous studies associated with CRP levels and/or with diabetes incidences, but not mediating the pathway between the association of CRP and diabetes incidences.

Effect modifiers were those that have been indicated interacting with hs-CRP but exhibiting conflicting conclusions in previous studies.

In paragraph 5 in Introduction on page 6, we have added the rationale for investigating interaction between hs-CRP and sex, hypertension, BMI or abdominal obesity due to conflicting conclusions in previous studies.

In addition, in Methods under "Potential confounders" and "Potential effect modifiers" on page 11, we'd described the various variables and included the references from which the rationales were based on.

The recruitment strategy could be more specific about how many people were selected in Troms 6 and Troms 7. The variable selection strategy can be more specific with rationale for the multivariable model.

Authors' response:

We thank Dr. Naheed for the insightful comment.

We have edited the Methods section under "Study sample" on pages 9-10 and had supplemented by Figure 2 - "study sample inclusion flow chart" to better describe the strategy in selecting the study sample.

We have further elaborated the strategy and rationale for selecting the variables (potential confounders and effect modifiers) for the multivariable model in the introduction section in paragraphs 4 & 6 on pages 6-7 and further illustrated the selection of confounders by Figure 1 - "the conceptual framework".

In brief, potential confounders were those that have been indicated in previous studies associated with CRP levels and/or with diabetes incidences, but not mediating the pathway between the association of CRP and diabetes incidences.

Effect modifiers were those that have been indicated interacting with hs-CRP but exhibiting conflicting conclusions in previous studies.

In the Methods section under "Variables in the analysis" on pages 10-13 has been re-structured for easier reading.

Why interaction analysis is required for this study should be explained in the rational section. What strategy did the authors use to define the confounding and interaction variables that needed to be discussed in the data analysis plan. The interaction analysis can be detailed and presented in a separate tables. The key outcome should be confirmed by sensitivity analysis to explore whether the

association between hs-CRP and incident of diabetes exist in a sub-group analysis in order to support discussion.

Authors' response:

We thank Dr. Naheed for the insightful comment.

In paragraph 5 in Introduction on page 6, we have added the rationale for investigating interaction between hs-CRP and sex, hypertension, BMI or abdominal obesity due to conflicting conclusions in previous studies.

In addition, in Methods under "Potential confounders" and "Potential effect modifiers" on page 11, we'd described the various variables and included the references from which the rationales were based on.

We have added to Table 2 on page 17, the analysis outcome of the interaction analysis with analysis methods described in Methods under "Evaluation of potential effect modification" on page 18. We have added additional details and references in the introduction in paragraphs 4 & 6 on pages 6-7 regarding the selection of the potential confounders that were based on previous studies.

In addition, we have added Figure 1 - the conceptual framework to illustrate the relationship of the selected variables (potential confounders or comorbidity) with hs-CRP (exposure) and diabetes (outcome).

In brief, from previous studies, potential confounders that have been indicated associated with CRP levels and/or with diabetes incidences, but not mediating the pathway between the association of CRP and diabetes incidence.

In Methods section, the "Statistical analysis" has been re-structured for easier reading.

The key outcome was re-analyzed by sensitivity analysis: Instead of analyzing hs-CRP as 3-quantiles, we had re-analyzed the regression model using log hs-CRP as a continuous independent variable. We presented the data as crude model and as multivariable adjusted model in Table 2 on page 17. We found that there were 28% higher odds of diabetes development for every one-log increase of hs-CRP.

In this study how the level of statistical significance was determined is unclear? Author should mention the cutoff of the probability value for the level of significance.

Authors' response:

We thank Dr. Naheed for the insightful comment.

Authors would prefer to present all P-values without emphasizing any arbitrary cutoff level of significance. Hence readers may be able to appreciate the impact of the CRP-diabetes association with the reported p-values, based on the context of the entire study.

How physical activity status was assessed in the study is unclear. It is preferable to use a globally accepted tool to measure physical activity rather than self-reported data

Authors' response:

We thank Dr. Naheed for the insightful comment.

Authors acknowledge the limitation of this retrospective population-based survey dataset. We, nevertheless, has added details that defined the levels (sedentary, low, moderate, vigorous) of physical activities from the original surveys in Methods under “Definition of the variables (confounders and/or effect modifiers)” on page 12.

In this study, author have used BMI (< 25 kg/m²) for normal weight rage which is not correct. The WHO criteria or any others globally acceptable cut off values can be adopted for the analysis.

Authors’ response:

We thank Dr. Naheed for the insightful comment.

We agree with reviewer that the BMI categories were not defined into 6 groups as recommended by WHO [1]:

[1] <https://www.who.int/europe/news-room/fact-sheets/item/a-healthy-lifestyle---who-recommendations>

WHO definition (kg/m ²)	Definition in current study (kg/m ²)
Underweight (< 18.5)	Normal (< 25.0)
Normal weight (18.5 - 24.9)	
Pre-obesity (25.0 - 29.9)	
Obesity class I (30.0 - 34.9)	
Obesity class II (35.0 - 39.9)	
Obesity class III (> 40.0)	Overweight (25.0 - 29.9)
	Obesity (≥ 30)

Norwegian Institute of Public Health based on the sixth survey of the Tromsø Study (Tromsø6) [2]:

Norwegian Institute of Public Health definition	Age standardized	
	2006 -2007, men (%)	2006 - 2007, women (%)
Overweight (≥ 25 kg/m ²)	51.2	38.0
Obesity (≥ 30 kg/m ²)	20.2	19.5
Overweight and obesity	71.4	57.5

[2] <https://www.fhi.no/en/op/Indicators-for-NCD/Overweight-and-obesity/overvekt-og-fedme-blant-voksne-indikator-14/>

[3] <https://www.fhi.no/en/op/hin/health-disease/overweight-and-obesity-in-norway---/>

According to Norwegian Institute of Public Health, as described above, there was high prevalence of overweight and obesity combined in Norway at the time of the baseline survey used in our current study. Our data also showed a high prevalence of 63.6% (Table 1, page 14) for the overweight and obesity combined.

In contrast to the high prevalence of overweight and obesity combined, the number of underweight (< 18.5 kg/m²) individuals was low with the rest of the 25% of men and 40% of women were categorized to normal BMI (18.5 - 24.9 kg/m²) according to Norwegian Institute of Pulic Health [3]. Due to data sparsity in the underweight group, we had combined the small number of underweight into the “normal BMI” group.

Similarly, due to data sparsity, we had collectively categorized individuals with ≥ 30 kg/m² as obesity, regardless whether they belonged to class I - III by WHO’s definition.

We had included this description in Methods under “Definition of the variables (confounders and/or effect modifiers)” on page 12.

We have added discussion regarding this under “Limitations of the study” on page 26.

The methodology lacks an appropriate reference for each cut off value for anthropometric and clinical assessment.

Authors’ response:

We thank Dr. Naheed for the insightful comment.

We have added reference and explanation for the values used in Methods under “Definition of the variables (confounders and/or effect modifiers)” on pages 12-13:

- Physical activity: defined from self-report as sedentary (reading, watching TV, or other sedentary activity), low (walking, cycling, or other forms of exercise at least 4 hours a week), moderate (recreational sports, heavy gardening, etc. at least 4 hours a week), or vigorous (hard training or sports competitions, regularly several times a week) [ref: 51 in the manuscript]
- BMI: was categorized as normal ($< 25 \text{ kg/m}^2$), overweight ($25 - 29.9 \text{ kg/m}^2$), or obesity ($\geq 30 \text{ kg/m}^2$). [ref 47 in the manuscript]. Due to data sparsity, the participants who were underweight ($< 18.5 \text{ kg/m}^2$) were combined into the normal BMI group.
- Abdominal obesity was categorized as abdominal obese when waist-to-height ratio ≥ 0.5 . It was suggested by previous systematic review and meta-analysis studies that a waist-to-height ratio cutoff of 0.5 was suitable for adults of both sex and different ethnic backgrounds [ref 27, 56 in the manuscript]
- Hypertension was defined by systolic blood pressure $\geq 140 \text{ mmHg}$ and/or diastolic blood pressure $\geq 90 \text{ mmHg}$, [ref 57 in the manuscript] and/or self-reported use of blood pressure lowering drugs (ATC-code C02, antihypertensives; C03, diuretics; C07, beta blocking agents; C08, calcium channel blockers; C09, renin-angiotensin acting agents) [ref 55 in the manuscript] for treating symptoms.

Why the hs-CRP level stratified into three categories should be justified and if there is any clinical implication of presenting stratified analysis for three tertile should be justified. The authors have stratified the hs-CRP data into three different tertial which transformed a continuous data into ordinal data.

Authors’ response:

We thank Dr. Naheed for the insightful comment.

This study did not aim to deduce a clinically relevant hs-CRP cutoff - rather to investigate whether the elevated baseline hs-CRP, a marker for systemic inflammation, associates with future diabetes cases.

Except for the upper threshold of $<10 \text{ mg/L}$ to exclude individuals with potential acute inflammation, the 3-quantiles of hs-CRP were not stratified by any reference to clinical cutoff. Conversely, the 3-quantiles were generated arbitrarily by stratifying the study sample into approximately 3 equal groups with ascending hs-CRP level from tertile 1 to tertile 3.

The idea was to first examine if there was dosage effect from exposure and whether future diabetes cases were associated with higher baseline hs-CRP (Table 2, page 17).

In the sensitivity analysis, authors re-analyzed the model by using log hs-CRP as a continuous independent variable and found similarly robust positive CRP-diabetes association (Table 2, page 17).

At present, authors do not aware of any clinical cutoff of hs-CRP being used for diabetes-related clinical practices.

hs-CRP level were log transformed due to non-normal structure. It is not mentioned why the authors did not follow a similar strategy for others variables (blood glucose, HDL cholesterol).

Authors' response:

We thank Dr. Naheed for the insightful comment.

Other continuous variables including diastolic blood pressure, systolic blood pressure, non-HDL cholesterol, and waist-height ratio were normally distributed, and therefore were not log-transformed.

We had added description in the Methods pertaining to these continuous variables whether normally or non-normally distributed in Methods under "Statistical analysis" subheading "Baseline characteristics" on page 13, and the types of statistical methods used in the analysis that the data was presented in Table 1 (pages 14-16). We also described how hs-CRP was first log transformed for the regression model in the sensitivity analysis on page 18.

Data interpretation and conclusion

Description of basic characteristics is unclear in the result section. A baseline difference between patients with and without diabetes is required to understand how the distribution of study participants differs according to the disease status. Age group has not been categorized in the consistent manner.

Authors' response:

We thank Dr. Naheed for the insightful comment.

We have excluded baseline diabetes in the study sample and therefore, it is not permissible to draw comparison with the absence of baseline diabetes patients.

We have, nevertheless, described baseline characteristics of all the included study sample (n=8,071) in Table 1 (pages 14-16) pertaining to their demographic and clinical attributes.

The majority of the participants were categorized in 5-year age groups except for the youngest (30 - 39 years) and the eldest (75 - 87 years) due to data sparsity. They account for 3.5% and 3.9%, respectively, of the total study sample (Table 1).

We had added this explanation in Methods under "Definition of the variables (confounders and/or effect modifiers)" on page 12.

In Table 1, the author should specify the type of statistical test applied to assess changes of mean and SD across tertile of hs-CRP level. P-value for trend can be appropriate in this case to report mean or median changes across tertiles of hs-CRP level.

Authors' response:

We thank Dr. Naheed for the insightful comment.

We have described the type of statistical test used in Table 1 in the footnote/caption to the Table 1 (pages 14-16):

“p-value across hs-CRP tertiles was obtained by Pearson’s chi-squared test for categorial variables, by Kruskal-Wallis test for continuous variables with non-normal distribution, or by one-way ANOVA for continuous variables with normal distribution”.

We have also included the type of statistical test used in Table 1 in Methods under “Statistical analysis” subheading “Baseline characteristics” on page 13.

In Table 2, the author used a logistic regression model to demonstrate the relationship between hs-CRP and diabetes incidence. It was unclear how the model was adjusted for the correlated structure of data (repeated measure of same individuals) and how the time difference was adjusted. The author can report an incidence rate ratio instead of odds ratio, which may provide better epidemiological insight into this analysis.

Authors' response:

We thank Dr. Naheed for the insightful comment.

Our study involved 2 population-based surveys at two discrete times: the baseline in Tromsø6 (2007-2008) and the follow-up time in Tromsø7 (2015-2016) after a 7-year interval. There was no data collection between the years of the two surveys.

The nature of the dataset is not permissible for analyses such as time-to-event or incidence rate ratio.

Authors proposed that logistic regression was more suitable in this instance.

Conclusion

Authors can interpret the findings in a separate chapter like discussion and conclusion.

The primary goal of the study was to investigate the association between hs-CRP and diabetes.

However, the current analysis is insufficient to justify whether the association is independent of others demographic, anthropometric and clinical variables. Authors should investigate the association between hs-CRP and diabetes incident in terms of sensitivity analysis.

Authors' response:

We thank Dr. Naheed for the insightful comment.

Authors apologies for the typo error on the headings of the 2 sections - we had thus corrected back to “Discussion” for pages 22-26 and followed by “Conclusion” on page 27.

In the current study, authors had demonstrated that the CRP-diabetes association was still significant after adjusting for a number of variables in the first regression model using hs-CRP tertiles and also in the sensitivity analysis using log hs-CRP as continuous variable in the second regression model with similar robustness.

In this instance, authors would very much like to seek reviewer's suggestions on any specific additional potential confounders and/or suggestions of any specific types of sensitivity tests in mind to further examine the CRP-diabetes association?

The authors can describe the findings contextualizing the key messages in terms of sociodemographic, clinical and anthropometric factors of diabetes patients

We have added contextualization of the findings in terms of sociodemographic, clinical and anthropometric factors to our conclusions (page 27)

In final conclusion, hs-CRP has been identified as a predictor of future diabetes development. The current analysis, however, does not support this statement, and the findings only show a positive association that needs to be confirmed by further sub-analysis.

The conclusion should be rewritten and revised in light of the study's objective and key messages.

Authors' response:

We thank Dr. Naheed for the insightful comment.

We have rephrased our title and objectives from "predictive value" to investigate "the association" of hs-CRP for future diabetes development - to better suit the entire study presented here.

Minor comments:

1. Author can remove P-value if report Odds ratio and 95% CI of Odds Ratio

Authors would like to keep the p-values in Table 1 to help readers to appreciate the difference of the baseline characteristics across the hs-CRP tertiles.

2. Caption of Table 1 is missing

Caption/footnotes of Table 1 is present at the bottom of the table (page 16). Authors had a concern whether perhaps there was any issue in the pdf during transmission to the reviewer?

3. In this analysis, cox regression can be applied by sequential model to estimate hazard ratio for the incident of diabetes after adjusted with different factors (clinical, anthropometric, medications)

Our study involved 2 population-based surveys at two discrete times: the baseline in Tromsø6 (2007-2008) and the follow-up time in Tromsø7 (2015-2016). There was no data collection during the years between the 2 surveys. The nature of the dataset is not permissible for time-to-event analysis such as cox regression model.

4. Please move this sentence into the data analysis part "the distribution of CRP was skewed and natural log-transform was used for the statistical analysis".

We had added the sentence in Methods section under "Variables in the analyses" and also under "Statistical analysis" on page 11 (1st paragraph) and page 18 (sensitivity analysis), respectively.

5. Please use the term "Multivariable" instead of "Multivariate" as because multivariable indicates an association between single response/outcome variable and multiple explanatory variables

Accordingly, we had changed all "Multivariate" to "multivariable".

Reviewer: 2

Dr. Olaitan Akinboboye, Medical College of Wisconsin

Comments to the Author:

The article's title is "The predictive value of C-reactive protein for future development of diabetes in a general population: The Tromsø Study 2007-2016." This study aimed to determine the predictive value of hs-CRP for diabetes development in a Norwegian general population.

It sounds interesting, primarily because of the sample population. However, a few concerns need to be addressed.

1. For the multiple regression analysis, please clarify the methods for variables selection into the model

Authors' response:

We thank Dr. Olaitan Akinboboye for the insightful comment.

In response to the reviewer, we have added in the text: (1) In introduction, in paragraphs 4-6 on page 6-7, (2) in Methods under sections "Variables in the analyses" on pages 11-13, and "statistical analysis" on pages 13 and 18, (3) and also "Figure 1 - conceptual framework" to clearly describe the rationale for selecting the variables as potential confounders and/or potential effect modifiers in the model, the definition of the variables, and how the variables were used in the model analyses.

In brief, potential confounders were those that have been indicated in previous studies associated with CRP levels and/or with diabetes incidences, but not mediating the pathway between the association of CRP and diabetes incidences.

Effect modifiers were those that have been indicated interacting with hs-CRP but exhibiting conflicting conclusions in previous studies.

Relevant references for the rationale of each variables were added to the text.

2. What informed the values (upper and lower limits) used in grouping the hs-CRP levels into tertiles?

Authors' response:

We thank Dr. Olaitan Akinboboye for the insightful comment.

Except for the upper threshold of <10 mg/L to exclude individuals with potential acute inflammation, the 3-quantiles of hs-CRP were not stratified by any reference to clinical cutoff. Conversely, the 3-quantiles were generated arbitrarily by stratifying the study sample into approximately 3 equal groups with ascending hs-CRP level from tertile 1 to tertile 3.

The idea was to first examine if there was dosage effect from the exposure and whether future diabetes cases were associated with higher baseline hs-CRP.

In the sensitivity analysis, authors re-analyzed the model by using log hs-CRP as a continuous independent variable and found similarly robust positive CRP-diabetes association (Table 2, page 17).

At present, authors do not aware of any clinical cutoff of hs-CRP being used for diabetes-related clinical practices.

3. How did the authors handle pre-diabetes individuals with HbA1c between 5.7% – 6.4%

Authors' response:

We thank Dr. Olaitan Akinboboye for the insightful comment.

Individuals deemed prediabetes, defined by HbA1c between 5.7 – 6.4% or 39 – 47 mmol/ml, were included in this current study.

Prediabetes state has been considered as risk for diabetes transformation. Although positive association had been previously implicated, the temporal relationship between elevated hs-CRP and prediabetes condition remained unclear. In addition, given that there is also certain extent of inaccuracies due to some glucose-independent variation in HbA1c measurement, authors considered little contradiction by including this group in the study sample.

We had included this discussion on page 24.

4. Does the Tromsø Study collect cancer information? If yes, did the authors consider controlling for cancer?

Authors' response:

We thank Dr. Olaitan Akinboboye for the insightful comment.

There was no cancer information at the baseline survey during 2007-2008.

Reviewer: 1

Competing interests of Reviewer: No competing interest

Reviewer: 2

Competing interests of Reviewer: No

1

VERSION 2 – REVIEW

REVIEWER	Naheed, Aliya International Centre for Diarrhoeal Disease Research Bangladesh, Initiative for Noncommunicable Diseases
REVIEW RETURNED	26-Jul-2023

GENERAL COMMENTS	I would like to thank the authors for addressing the comments in details. While most of the responses are satisfactorily addressed there are two issues that would need further clarity for an honest the interpretations of the results presented in the revised manuscript. Please find my observations and suggestions below: - My comment: The rationale of excluding individuals with hs-CRP level $\geq 10\text{mg/L}$ indicating an acute inflammatory condition was not understood, since an acute exacerbation of a chronic inflammation might be an important predictor for diabetes. The rationale of the study should be well supported by robust scientific evidence.
---

	Authors response: Since acute inflammation may resolve after a short period of time and may not necessarily always transit subsequently into chronic inflammation, we have a concern that by including hs-CRP level $\geq 10\text{mg/L}$, an indication of acute inflammatory condition, may contribute to the risk of measurement error. My feedback on authors' responses: Justifications provided by the authors are not well supported by sufficient evidence. The authors should present a brief analysis with the individuals excluded from the study including essential parameters and compare with the individuals recruited in the study to indicate what parameters would support authors' argument in favor of potential measurement of errors in order to establish a strong justification for excluding individuals with a hs-CRP level $\geq 10\text{mg/L}$. The authors are requested to remove the cases of self-reported diabetes who did not have any supporting evidence. Self-reported diabetes is not a reliable measure of diabetes, and including these cases in the analysis could lead to misleading results. - My comment: In this study, authors have used BMI ($< 25 \text{ kg/m}^2$) for normal weight range which is not correct. The WHO criteria or any others globally acceptable cut off values can be adopted for the analysis. Authors response: In contrast to the high prevalence of overweight and obesity combined, the number of underweight ($< 18.5 \text{ kg/m}^2$) individuals was low with the rest of the 25% of men and 40% of women were categorized to normal BMI ($18.5 - 24.9 \text{ kg/m}^2$) according to Norwegian Institute of Public Health [3]. Due to data sparsity in the underweight group, we had combined the small number of underweight into the "normal BMI" group. My feedback on authors' responses: BMI $< 18.5 \text{ kg/m}^2$ indicates underweight and it cannot be interpreted as normal BMI. The authors either should create a 'underweight' group for a proper analysis or exclude the underweight group from the analyses, if the number of study participants are very small.
--	--

REVIEWER	Akinboboye, Olaitan Medical College of Wisconsin, Institute of Health and Equity
REVIEW RETURNED	09-Apr-2023
GENERAL COMMENTS	The authors have provided sufficient response to my comments.

VERSION 2 – AUTHOR RESPONSE

We are very grateful for the reviewers' comments and the opportunity to respond to them.

Our response is given below in blue:

Reviewer: 2

Dr. Olaitan Akinboboye, Medical College of Wisconsin

Comments to the Author:

The authors have provided sufficient response to my comments.

Authors' response:

We would very much like to thank Dr. Olaitan Akinboboye for his kind appreciation of our revision. We have had found his review of our manuscript very constructive and insightful.

Reviewer: 1

Dr. Aliya Naheed, International Centre for Diarrhoeal Disease Research Bangladesh

Comments to the Author:

I would like to thank the authors for addressing the comments in details. While most of the responses are satisfactorily addressed there are two issues that would need further clarity for an honest the interpretations of the results presented in the revised manuscript. Please find my observations and suggestions below:

- My comment: The rationale of excluding individuals with hs-CRP level $\geq 10\text{mg/L}$ indicating an acute inflammatory condition was not understood, since an acute exacerbation of a chronic inflammation might be an important predictor for diabetes. The rationale of the study should be well supported by robust scientific evidence.

Authors response: Since acute inflammation may resolve after a short period of time and may not necessarily always transit subsequently into chronic inflammation, we have a concern that by including hs-CRP level $\geq 10\text{mg/L}$, an indication of acute inflammatory condition, may contribute to the risk of measurement error.

My feedback on authors' responses: Justifications provided by the authors are not well supported by sufficient evidence. The authors should present a brief analysis with the individuals excluded from the study including essential parameters and compare with the individuals recruited in the study to indicate what parameters would support authors' argument in favor of potential measurement of errors in order to establish a strong justification for excluding individuals with a hs-CRP level $\geq 10\text{mg/L}$.

Authors' response:

Authors thank Dr. Aliya Naheed for the constructive comments.

Based on the evidences presented in the literature of previous clinical and laboratory studies (as explained further below), the authors had excluded participants with hs-CRP level $\geq 10\text{mg/L}$, an indication of acute inflammatory condition, to avoid the risk of measurement error of exposure.

The World Health Organization reported that serum CRP level is normally less than 10mg/L and hence can be utilized as an early biomarker for detecting inflammation or infection [Ref:1].

During severe viral infection, typical CRP level in serum may range between $10\text{-}40\text{ mg/L}$. Serum CRP level may reach $40\text{-}200\text{ mg/L}$ during active inflammation and bacterial infection, while a level of $200\text{-}400\text{ mg/L}$ may be observed for severe bacterial infection and burns [Ref:2].

During the clinical course of acute disease states or infections, CRP levels could rise rapidly in early hours and peak around 48 hours, after which the level might gradually subside according to the speed of resolution of the disease states or infections [Ref:3].

Other studies had found that patients with acute appendicitis exhibited CRP level >12mg/L [Ref:4,5].

Serum CRP level was deemed useful in monitoring clinical changes of diseases. Previous clinical studies considered a CRP level less than 10mg/L deemed indisputably negative for bacterial meningitis [Ref:6].

Authors had included these six references [Ref 53-58 in manuscript] in the Methods section under “Study sample” on p.9 of the revised manuscript, citing these previous reports.

We are concerned that including people with hs-CRP >10mg/L on the day of the examination would lead to misclassification of the exposure for people whose CRP is transiently raised due to acute infection (most likely resulting in bias towards the null) as we would be unable from one measurement to distinguish inflammation due to acute infection from chronic conditions. For this reason, we are unwilling to include them.

To examine this, authors added additional sensitivity analyses by including study participants with CRP \geq 10mg/L as described in table 3 on p.19. Indeed, authors found there was a light bias towards null, especially when study sample was analyzed as 3-quantiles of hs-CRP. Nevertheless, the CRP-diabetes positive association remained robust either analyzing hs-CRP as 3-quantiles or as continuous variables. Description of the additional methods, results, and discussion were added on p.18, pp.22-23, and p.24, respectively.

References:

[1] C-reactive protein concentrations as a marker of inflammation or infection for interpreting biomarkers of micronutrient status. World Health Organization. 2014.

Available: <https://apps.who.int/iris/handle/10665/133708> [Accessed 6 Aug 2023].

[2] Stuart J, Whicher JT. Tests for detecting and monitoring the acute phase response. *Arch Dis Child* 1988;63:115-7.

[3] Clyne B, Olshaker JS. The C-reactive protein. *J Emerg Med* 1999;17:1019-25.

[4] Marchand A, Van Lente F, Galen RS. The assessment of laboratory tests in the diagnosis of acute appendicitis. *Am J Clin Pathol* 1983;80:369-74.

[5] van Dieijen-Visser MP, Go PMNYH, Brombacher PJ. The value of laboratory tests in patients suspected of acute appendicitis. *Eur J Clin Chem Clin Biochem* 1991;29:749-52.

[6] Peltola HO. C-reactive protein for rapid monitoring of infections of the central nervous system. *Lancet* 1982;319:980-3.

The authors are requested to remove the cases of self-reported diabetes who did not have any supporting evidence. Self-reported diabetes is not a reliable measure of diabetes, and including these cases in the analysis could lead to misleading results.

Authors' response:

Authors thank Dr. Aliya Naheed for the constructive comments.

The Tromsø Study consists of a population-based survey dataset **with the support of concurrent lab-based measurements including the lab-based measure of glycated hemoglobin (HbA1c) levels**. While we agree within a clinical trial or clinical practice self-reported diabetes alone would not be used, within epidemiological population-based studies this is a standard practice. Within the manuscript, we have included several papers about the validity of self-reported diabetes (re 74-76).

The authors consider that many diabetes patients would be usually be able to control their diabetes through diet and/or medication. If one excludes the use of self-reported cases and only consider those found on the day of the survey was taken from clinically high measurement of HbA1c, the identified diabetes cases potentially could have been restricted mostly to a specific group of severe cases with poor control by diet and/or medication - this would potentially lead to a biased subset of cases.

To capture known and unknown diabetes cases, authors used (1) self-reported diabetes, (2) self-reported medication for diabetes for treating symptoms, **and also (3) a more robust lab-based blood measurement of HbA1c** to support both the data of self-reported diabetes and diabetes medication use for treating symptoms. **The dataset has high frequency of lab-based measurement of HbA1c** to complement the self-reported data. **Together with the use of lab-based blood test of HbA1c levels of participants**, authors believe that would increase the sensitivity and specificity of the diabetes case definition.

This case definition strategy would also help to identify the known and unknown/undiagnosed diabetes cases where the participants had no prior knowledge of their diabetes status, as previously studied and published [Ref 54. Langholz PL, Wilsgaard T, Njølstad I, *et al.* Trends in known and undiagnosed diabetes, HbA1c levels, cardiometabolic risk factors and diabetes treatment target achievement in repeated cross-sectional surveys: the population-based Tromsø Study 1994-2016. *BMJ Open* 2021;11:e041846].

The authors have included these details in the first revision regarding the definition of diabetes case in Methods on page 11 under "Outcome Definition (Diabetes)" and had also discussed on pp.27-28 under the "limitations of the study" regarding the limitations of the use of self-reported diabetes and self-reported diabetes medication for treating symptoms. Authors have also included discussion on how the additional use of lab-based measurement of HbA1c would improve the sensitivity and specificity in identifying true diabetes cases among the study samples in the study.

On the other hand, authors had also discussed previous studies [Ref 74-76] that had found high specificity, especially among urban dwellers, in self-reported diabetes.

For reviewer's convenience, authors have colored text all above-mentioned description on p.11 & pp.27-28 in the current second revision.

- My comment: In this study, author have used BMI (< 25 kg/m²) for normal weight range which is not

correct. The WHO criteria or any others globally acceptable cut off values can be adopted for the analysis.

Authors response: In contrast to the high prevalence of overweight and obesity combined, the number of underweight ($< 18.5 \text{ kg/m}^2$) individuals was low with the rest of the 25% of men and 40% of women were categorized to normal BMI ($18.5 - 24.9 \text{ kg/m}^2$) according to Norwegian Institute of Public Health [3]. Due to data sparsity in the underweight group, we had combined the small number of underweight into the “normal BMI” group.

My feedback on authors' responses: BMI $< 18.5 \text{ kg/m}^2$ indicates underweight and it cannot be interpreted as normal BMI. The authors either should create a 'underweight' group for a proper analysis or exclude the underweight group from the analyses, if the number of study participants are very small.

Authors' response:

Authors thank Dr. Aliya Naheed for the constructive comments.

To better describe the included participants and to avoid unnecessary confusion, authors have eliminated the labeling of “normal” BMI pertaining to participants who had BMI $< 25 \text{ kg/m}^2$. Henceforth, the 3 new BMI categories in this second revision, in Methods (p.12), Table 1 (pp.14-16), and in Discussion (p.28) become participants with: $< 25 \text{ kg/m}^2$, $25 - 29.9 \text{ kg/m}^2$, or $\geq 30 \text{ kg/m}^2$. Indeed, among the total participants in either Tromsø6 and Tromsø7, there were very low frequency of up to 1% of women or men who had BMI less than 18.5 kg/m^2 or underweight according to WHO definition [Ref 78: Løvsløtten O, Jacobsen BK, Grimsgaard S, *et al.* Prevalence of general and abdominal obesity in 2015-2016 and 8-year longitudinal weight and waist circumference changes in adult and elderly: the Tromsø Study. *BMJ Open* 2020;10:e038465]. We had included this reference to Discussion on p.28.

We should also note that the dataset as received by the authors from the Tromsø Study included body mass index as a pre-categorized variable, so we were not able to separate out those who were less than 18.5 kg/m^2 within our dataset and exclude these people or analyze them as a separate group.